JCB Journal of Cell Biology

## REPORT

# REC drives recombination to repair double-strand breaks in animal mtDNA

Anna Klucnika[1,2], Peiqiang Mu[1,2,3], Jan Jezek[1,2] [ORCID], Matthew McCormack[1,2] [ORCID], Ying Di[1,2], Charles R. Bradshaw[1] [ORCID], and Hansong Ma[1,2] [ORCID]

**Mechanisms that safeguard mitochondrial DNA (mtDNA) limit the accumulation of mutations linked to mitochondrial and age-related diseases. Yet, pathways that repair double-strand breaks (DSBs) in animal mitochondria are poorly understood. By performing a candidate screen for mtDNA repair proteins, we identify that REC—an MCM helicase that drives meiotic recombination in the nucleus—also localizes to mitochondria in *Drosophila*. We show that REC repairs mtDNA DSBs by homologous recombination in somatic and germline tissues. Moreover, REC prevents age-associated mtDNA mutations. We further show that MCM8, the human ortholog of REC, also localizes to mitochondria and limits the accumulation of mtDNA mutations. This study provides mechanistic insight into animal mtDNA recombination and demonstrates its importance in safeguarding mtDNA during ageing and evolution.**

## Introduction

Preserving the integrity of the mitochondrial genome during development and ageing is crucial because the accumulation of mitochondrial DNA (mtDNA) mutations is linked to mitochondrial diseases (Bernardino Gomes et al., 2021), cancers (Gammage and Frezza, 2019) and age-related conditions (Monzio Compagnoni et al., 2020). Research in the past decade has begun to unravel mechanisms that correct single-stranded lesions in animal mitochondria, such as base excision repair (Kazak et al., 2012; Fu et al., 2020). Yet, mechanisms that repair double-strand breaks (DSBs) remain unclear. Homologous recombination (HR) occurs frequently in plant and yeast mitochondria to facilitate mtDNA DSB repair, replication, and segregation (Chevigny et al., 2020; Solieri, 2010). In contrast, mtDNA recombination in animals appears to be rare (Hagström et al., 2014; Hayashi et al., 1985). Nevertheless, reports of naturally occurring mtDNA recombinants in various animals species—including humans—suggest that animal mtDNA can undergo recombination (Ladoukakis and Zouros, 2001; Hoarau et al., 2002; D'Aurelio et al., 2004; Sato et al., 2005; Guo et al., 2006; Ujvari et al., 2007; Fan et al., 2012; Ma and O'Farrell, 2015; Strakova et al., 2016), although the significance of HR in safeguarding animal mtDNA remains unknown.

HR in the nucleus relies on the RecA-related recombinases RAD51 or DMC1. Recombination in *Arabidopsis thaliana* mitochondria also requires RecA-related recombinases (Miller-Messmer et al., 2012; Shedge et al., 2007). In *Saccharomyces cerevisiae*, mtDNA recombination is mediated by a RAD52-related protein, MGM101, independent of a RecA homolog

(Kleff et al., 1992; Mbantenkhu et al., 2011). While no RAD52-related proteins have been found in animal mitochondria, RAD51 and some other mediators of nuclear HR are reported to be mitochondrial in cultured mammalian cells (Dahal et al., 2018; Li et al., 2019; Luzwick et al., 2021; Mishra et al., 2018; Sage et al., 2010). It was even demonstrated that loss of RAD51 or related proteins results in mtDNA depletion upon oxidative stress (Sage et al., 2010). However, the mechanism of animal mtDNA recombination remains unknown as no protein has been shown to mediate homology-dependent repair in mitochondria.

Previously we showed that mtDNA HR occurs in *Drosophila* carrying two mitochondrial genotypes (Ma and O'Farrell, 2015). Spontaneous recombination was only detected at a low frequency, but induction of DSBs by expressing mitochondrially targeted restriction enzymes (mito-REs) greatly increased mtDNA recombination. Expression of mito-REs also allows recombinant mtDNA to be easily detected because it selects against the parental mitochondrial genomes (Ma and O'Farrell, 2015). This provides an in vivo system to investigate the mechanism of animal mtDNA recombination and to evaluate its impact on mtDNA integrity and animal physiology during development and ageing.

Here, we report an unexpected function of a meiotic helicase, REC/MCM8, in mtDNA repair and maintenance. By performing a candidate screen to identify mtDNA repair proteins, we found that REC is a dual-targeted protein that localizes to both the nucleus and mitochondria. The mitochondrial enrichment of REC was confirmed by endogenous tagging, subcellular

[1]Wellcome/Cancer Research UK Gurdon Institute, Cambridge, UK;   [2]Department of Genetics, University of Cambridge, Cambridge, UK;   [3]Guangdong Provincial Key Laboratory of Protein Function and Regulation in Agricultural Organisms, South China Agricultural University, Guangzhou, Guangdong, China.

Correspondence to Hansong Ma: hm555@cam.ac.uk.



fractionation, and proteinase K protection assays. We generated mutants that deplete only mitochondrial REC or knock out REC entirely and performed in vivo mtDNA recombination assay to show that REC is not essential for mtDNA replication, but functions to promote mtDNA HR and DSB repair in somatic and germline tissues. Additionally, loss of REC increased age-induced mitochondrial mutations and dysfunction. Finally, we show that MCM8, the human ortholog of REC, co-localizes with mtDNA and that MCM8 mutant cells accumulated more mtDNA mutations. This study provides the first demonstration of a protein that functions in mtDNA recombination and DSB repair in animals.

## Results and discussion

The characterized mtDNA repair pathways, such as base excision repair, are largely mediated by dual-targeted nuclear repair enzymes that are essential for mtDNA maintenance, independent of their functions in the nucleus (Nilsen et al., 1997; Simsek et al., 2011). We reasoned that this could also be the case for mtDNA recombination machinery. We endogenously tagged homologs of four nuclear recombination proteins that have been previously reported to localize to mitochondria of mammalian cells—RAD51 (Spn-B and Spn-D), MRE11, and NBS (Dahal et al., 2018; Li et al., 2019; Luzwick et al., 2021; Mishra et al., 2018; Sage et al., 2010; Wisnovsky et al., 2016)—and examined their subcellular localization in *Drosophila* ovaries by confocal microscopy. However, none showed mitochondrial enrichment in *Drosophila* oocytes (Fig. S1 A).

To identify novel DNA repair proteins that localize to mitochondria, we performed a co-localization screen by tagging 39 nuclear repair proteins with GFP and overexpressing these constructs in *Drosophila* S2R+ cells (Fig. 1 A and Table S1). Three proteins showed clear mitochondrial enrichment: REC and HDM, which are components of the meiotic recombination machinery, and XLF1 (CG12728), which mediates non-homologous end joining (Fig. 1 B; Ahnesorg et al., 2006; Blanton et al., 2005; Joyce et al., 2009; Matsubayashi and Yamamoto, 2003). We endogenously tagged these candidates with HaloTag to examine their subcellular localization in *Drosophila* ovaries and found that HDM levels were too low to detect and XLF1 only showed a modest mitochondrial signal (Fig. S1 B). Therefore, we focus here on REC, which showed the strongest mitochondrial signal in vivo (Fig. S1 C).

REC is a minichromosome maintenance (MCM) helicase that drives the formation of most meiotic crossovers in *Drosophila* ovaries by facilitating repair-specific DNA synthesis (Blanton et al., 2005; Matsubayashi and Yamamoto, 2003). Reflecting this function, REC was enriched in the oocyte nucleus from the early germarium to stages 5/6 before the transition from mid to late meiotic prophase I (Hughes et al., 2018; Fig. 1, C and D; and Fig. S1 C). Throughout oogenesis, REC also co-localizes with mitochondria of the oocyte and nurse cells (Fig. 1 D and Fig. S1 C). The nuclear and mitochondrial enrichment of REC was confirmed by blotting subcellular fractionations of ovaries (Fig. 1 E). To further investigate the localization of REC in mitochondria, we conducted a proteinase K protection assay

on freshly isolated crude mitochondria. Unlike Porin and Opa1, which are outer membrane and inner membrane proteins, respectively, mitochondrial REC is resistant to proteolytic degradation after the outer membrane integrity was compromised by hypo-osmotic treatment, similar to mitochondrial matrix proteins mtDNA polymerase PolG1 and ATP5A (Fig. 1 F). This indicates that REC localizes to the mitochondrial matrix.

The first 44 amino acids of REC are predicted by MitoProt to contain a mitochondrial-targeting signal (MTS; Claros and Vincens, 1996). Deletion of the putative MTS in endogenously tagged flies ($REC^{\Delta MTS}$-*Halo*) greatly reduced mitochondrial REC without affecting the amount of REC in the oocyte nucleus (Fig. 1, D and E; and Fig. S1, D and E). Nevertheless, a small fraction of REC remained in the mitochondria of $REC^{\Delta MTS}$-*Halo* (Fig. 1 E), indicating that there may be other signals contributing to the mitochondrial localization of REC. Although REC is not required for mitotic recombination (Matsubayashi and Yamamoto, 2003), it is also expressed in somatic tissues, but at a much lower level compared to ovarian tissues (Fig. 1 G and Fig. S1 F).

The human ortholog of REC is MCM8, which forms a complex with MCM9 to mediate both meiotic and mitotic recombination (Huang et al., 2020; Hustedt et al., 2019; Park et al., 2013). Unlike the replicative helicase MCM2-7, MCM8 is not essential for nuclear DNA replication but could be required for efficient replication elongation (Park et al., 2013). To test whether REC is required for mtDNA replication, we generated a null mutant line $REC^{KO}$-*Halo* (Fig. 1 D; and Fig. S1, D and E). Knocking out or overexpressing REC did not affect mtDNA copy number in eggs, or young and aged somatic tissues (Fig. 2 A and Fig. S2 A). $REC^{\Delta MTS}$ and $REC^{KO}$ mutants also did not show reduced lifespan or female fertility at 25°C (Fig. 2, B and C). Therefore, we conclude that, unlike the replicative helicase Twinkle (Milenkovic et al., 2013), REC is not essential for mtDNA replication.

Given REC's nuclear role in mediating meiotic recombination (Blanton et al., 2005; Grell, 1984; Matsubayashi and Yamamoto, 2003), we investigated whether mitochondrial REC plays a similar role in mtDNA recombination and DSB repair. Bleomycin is known to cause DSBs in both nuclear and mitochondrial DNA (Lim and Neims, 1987; Morel et al., 2008). By immunofluorescence staining, we found that the amount of REC in ovary mitochondria increased after 2 h bleomycin treatments (Fig. 3 A). Blotting subcellular fractionations of ovarian tissues also showed that REC is 1.49 and 1.16 times as abundant in mitochondrial and nuclear fractions of bleomycin-treated ovaries compared to untreated ones (Fig. 3 B). The increased REC in the two compartments could be due to increased translation and/or transcription as we detected a minor but insignificant increase in *rec* mRNA level after bleomycin treatment (Fig. S2 B). No change in mitochondrial REC was observed after treatment with oxidizing or cross-linking agents that cause other types of DNA damage (Fig. 3 A), suggesting that REC is particularly sensitive to DSBs in mtDNA.

We then performed an in vivo recombination assay by expressing mito-REs in heteroplasmic flies to measure the mtDNA recombination frequency in the presence and absence of REC. We created two heteroplasmic fly lines carrying

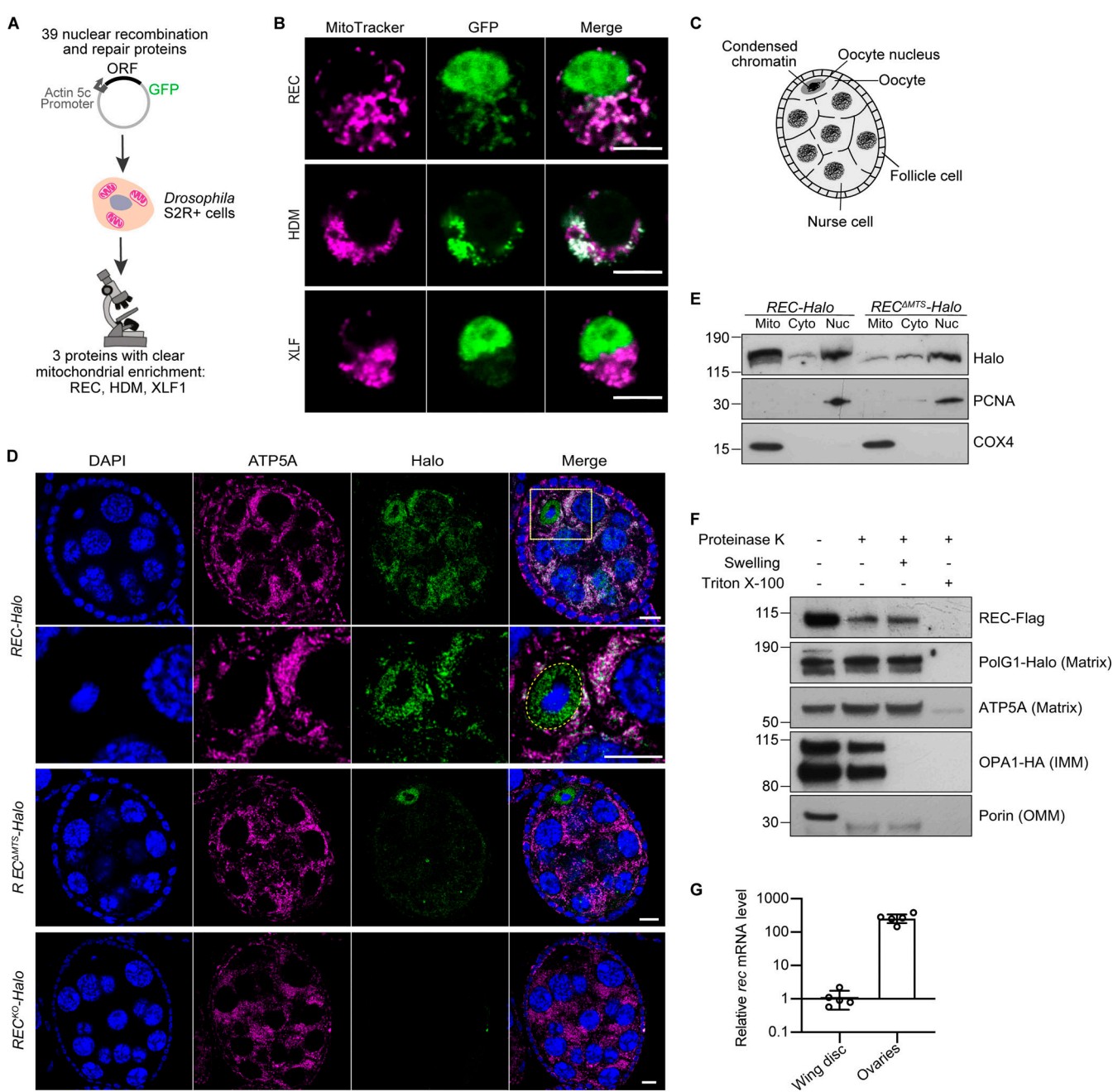

Figure 1. **A candidate screen identified the meiotic helicase REC as a dual-targeted protein that localizes to both the nucleus and mitochondria. (A)** Schematic of the candidate screen in *Drosophila* S2R+ cells. **(B)** Representative images of REC, HDM, and XLF1 tagged with GFP when overexpressed in S2R+ cells stained with MitoTracker to label mitochondria. Scale bar: 5 µm. **(C)** Diagram of a *Drosophila* stage 5/6 egg chamber. The oocyte and 15 nurse cells are surrounded by a single layer of somatic follicle cells. **(D)** Endogenous REC tagged with Halo co-localizes with the mitochondrial network (anti-ATP5A) of nurse cells and oocytes, and with oocyte nuclei that are highlighted by yellow-dashed lines (upper panel with zoomed-in views; a zoomed-out image of the same egg chamber is shown in Fig. S1 C as part of an ovariole that contains the germarium up to stage 9 egg chambers). $REC^{\Delta MTS}$-Halo ovaries showed reduced mitochondrial REC and unaltered nuclear REC in oocytes (middle panel). $REC^{KO}$-Halo ovaries, which carry a 130 bp deletion that introduces an early stop codon in *rec*, showed no Halo labeling (lower panel, Fig. S1, D and E). Scale bar: 10 µm. **(E)** Immunoblot of mitochondrial (Mito, anti-COX4), cytoplasmic (Cyto) and nuclear (Nuc, anti-PCNA) fractions of *REC-Halo* and $REC^{\Delta MTS}$-Halo eggs. This confirmed that $REC^{\Delta MTS}$-Halo flies have reduced mitochondrial REC, whereas the nuclear REC level remains unchanged. **(F)** Proteinase K protection assay showed that REC in the crude mitochondrial fraction was resistant to digestion after the outer mitochondrial membrane (OMM) integrity was disrupted by hypo-osmotic treatment. Porin, Opa1 (endogenously tagged with HA), PolG1 (endogenously tagged with Halo), and ATP5A were probed as the outer mitochondrial membrane (OMM), inner mitochondrial membrane (IMM) and matrix marker(s), respectively. **(G)** Relative *rec* mRNA levels measured by RT-qPCR, normalized to *ef1a* (n = 5 biological replicates; see Fig. S1 F for Ct values). Data represent mean ± SD. Source data are available for this figure: SourceData F1.

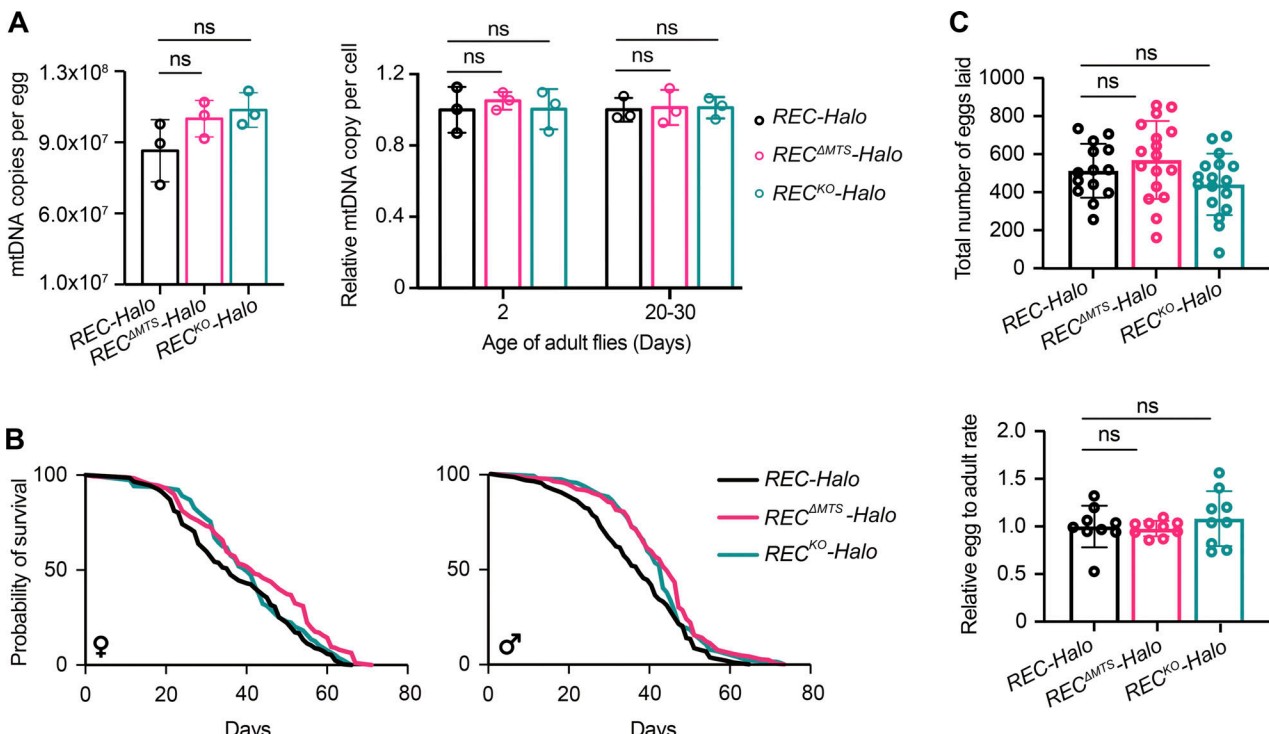

Figure 2. **REC has little impact on mtDNA copy number, longevity, or female fertility at 25°C. (A)** mtDNA copy numbers per egg or relative mtDNA copy numbers per adult head from flies of different ages measured by qPCR for *REC-Halo*, *REC^ΔMTS^-Halo*, and *REC^KO^-Halo*. Data represent mean ± SD, one-way ANOVA test with Dunnett's multiple comparisons; ns, P > 0.05 (*n* = 3 biological replicates). **(B)** Lifespan of *REC-Halo*, *REC^ΔMTS^-Halo*, and *REC^KO^-Halo*. *n* > 200 animals across two independent experiments for each genotype. **(C)** The number of eggs laid by each female over the first 20 d (*n* = 14–17 biological replicates) and the egg-to-adult (hatching and eclosing) rate over the first 10 d for *REC-Halo*, *REC^ΔMTS^-Halo*, and *REC^KO^-Halo* flies (*n* = 9 biological replicates). Data represent mean ± SD, one-way ANOVA test with Dunnett's multiple comparisons; ns, P > 0.05.

mitochondrial genotypes from closely related *Drosophila* species, which share enough sequence homology in their mitochondrial genomes to allow recombination, but also contain sufficient polymorphisms to allow identification and mapping of recombinant mtDNA. We then expressed mito-NciI in their germline to induce DSBs at different positions of the co-existing mtDNA genotypes (Fig. 3 C and Fig. S2 C). In *REC-Halo* flies, the DSB in mtDNA of one genotype was repaired using the homologous sequence on mtDNA of the other genotype, giving rise to progeny carrying recombinant mtDNA. All the recombinants isolated were the result of accurate homology-dependent repair (Fig. S2 D). The frequency of isolating progeny with recombinant mtDNA was reduced by ~40% in *REC^ΔMTS^-Halo* flies for both heteroplasmic lines. The frequency was further reduced by >80% of the control level in flies completely lacking REC (*REC^KO^-Halo*; Fig. 3 C). Notably, heteroplasmic *REC^ΔMTS^* or *REC^KO^* females produced a similar number of adult progeny as controls when mito-NciI was not expressed (Fig. S2 E). Furthermore, expression of mito-NciI in *REC^ΔMTS^* and *REC^KO^* flies homoplasmic for mtDNA that is resistant to NciI cutting (*mt:NciI^resistant^*) does not impair female fertility (Fig. S2 E). Therefore, the reduced frequency of isolating progeny with recombinant mtDNA in our recombination assay is due to impaired mtDNA repair in *REC^KO^* and *REC^ΔMTS^* flies. This demonstrates that REC facilitates germline mtDNA recombination.

We then examined the function of REC in mtDNA DSB repair in the soma. The expression of mito-REs in somatic tissues results in lethality of flies with wild-type mtDNA. This lethality can be rescued by mtDNA DSB repair in heteroplasmic flies (Ma and O'Farrell, 2015). Indeed, we observed a ~50% rescue in heteroplasmic *REC-Halo* flies after expressing mito-NciI under *nubbin-GAL4* in embryonic neuroblasts, ganglion mother cells, and larval wing discs (Fig. 3 D and Fig. S2 F). Reducing mitochondrial REC (*REC^ΔMTS^-Halo*) did not impact rescue in the soma (Fig. 3 C). It is possible that mtDNA repair mediated by the residual REC in mitochondria of *REC^ΔMTS^-Halo* flies (observed in Fig. 1 E and Fig. S1 E) is sufficient for a full rescue. However, the rescue was almost completely abolished in *REC^KO^-Halo* flies (Fig. 3 D). The failed rescue in *REC^KO^-Halo* flies was not due to impaired nuclear repair because expression of mito-NciI in flies homoplasmic for *mt:NciI^resistant^* caused no lethality (Fig. 3 D). Altogether, these data show that REC facilitates mtDNA DSB repair by recombination in both germline and somatic tissues.

Next, we tested whether REC also facilitates spontaneous mtDNA recombination without introducing DSBs by expressing mito-REs. Previously, we created flies heteroplasmic for *mt:ATP6[1]* and a temperature-sensitive (ts) lethal mutant genome that carries a point mutation in *mt:CoI*. Flies homoplasmic for *mt:ATP6[1]* are healthier and more robust than ts flies, but polymorphisms in the non-coding region of the ts genome grant it a selfish transmission advantage so that it displaced *mt:ATP6[1]*

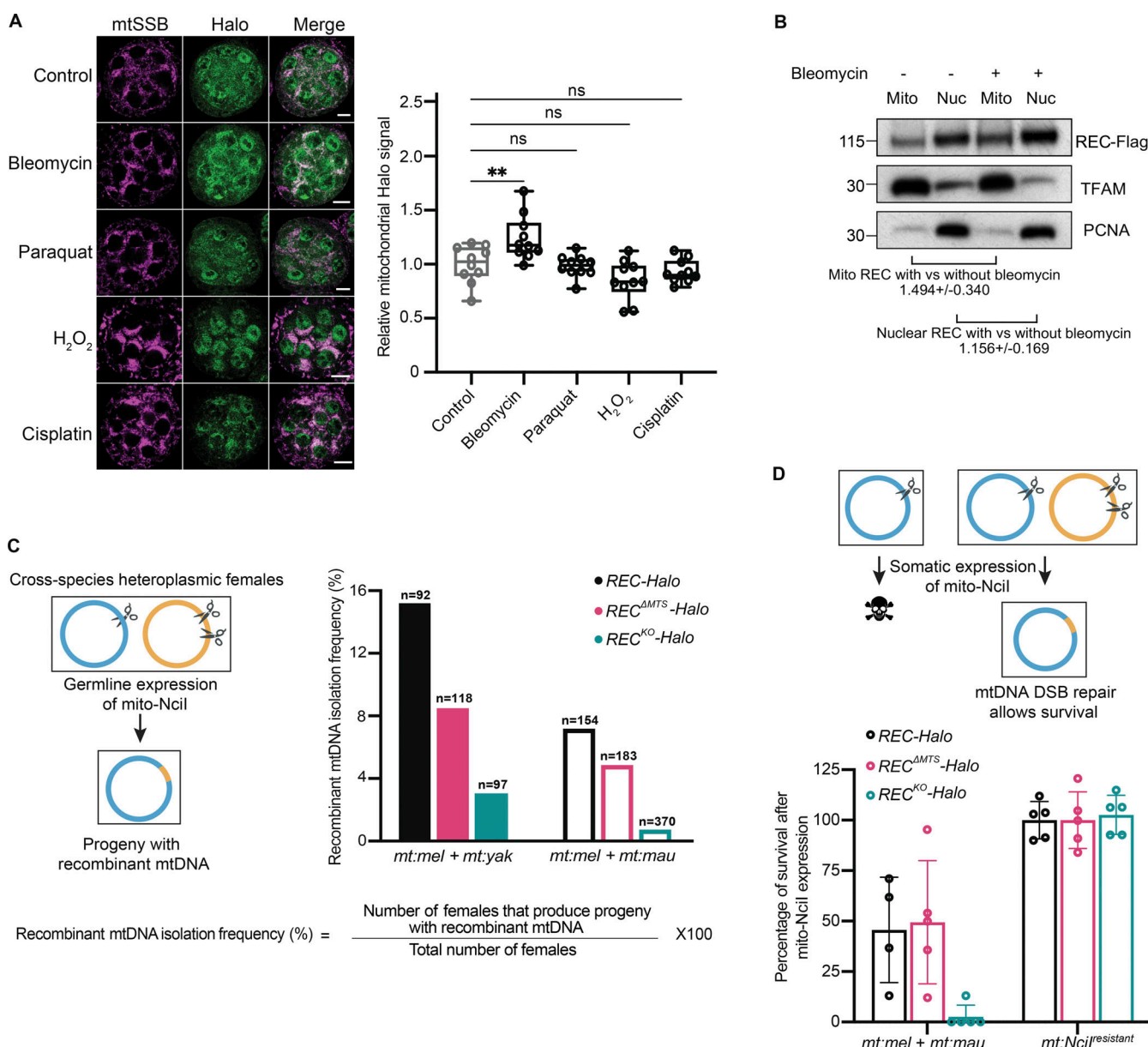

**Figure 3. REC mediates recombination-based mtDNA repair upon the introduction of DSBs. (A)** Representative images and quantification of Halo signals in *Ubi-mtSSB-GFP*; *REC-Halo* ovaries after treatment with DNA-damaging agents for 2 h (*n* = 10 biological replicates). Scale bar: 10 µm. Data represent mean ± SD. One-way ANOVA with Dunnett's multiple comparisons. The average intensities of Halo in mitochondrial regions defined by mtSSB-GFP signal were measured. **(B)** Representative immunoblot images of REC (endogenously tagged with Flag) in mitochondrial and nuclear fractions of ovaries with and without bleomycin treatment. The amount of REC was normalized to the amount of TFAM or PCNA in the same sample by quantifying band intensity. Three experiments were performed (see SourceDataF3). Data represent mean ± SD. **(C)** Schematic of germline recombination assay and subsequent recombinant mtDNA isolation frequency as a percentage of the number of flies examined (*n* = 97–370 biological replicates). Parental flies were heteroplasmic for *D. melanogaster* and *Drosophila yakuba* mtDNA (*mt:mel* + *mt:yak*, ~94% mtDNA sequence homology, filled bars) or *D. melanogaster* and *Drosophila mauritiana* mtDNA (*mt:mel* + *mt:mau*, ~95% mtDNA sequence homology, hollow bars). *D. melanogaster* mtDNA contains a single NciI recognition site at 3,648 bp. *D. yakuba* and *D. mauritiana* mtDNA contain two NciI recognition sites at 2,213 + 4,526 bp and 3,612 + 7,223 bp, respectively. The resulting progeny after expressing mito-NciI driven by *nos-GAL4* (germline) were homoplasmic for a certain recombinant mitochondrial genotype that lacks the NciI recognition sites (Fig. S2 D). The frequency of isolating recombinant mtDNA is measured by the percentage of heteroplasmic females expressing mito-NciI in germline that produced progeny with recombinant mtDNA. **(D)** Schematic of somatic recombination assay and subsequent rescue frequency. The rescue was calculated as the percentage of adult progeny that express mito-NciI versus their siblings that did not express mito-NciI (Fig. S2 F; *n* = 4–5 crosses). Data represent mean ± SD. Source data are available for this figure: SourceData F3.

over a few generations even at the restrictive temperature (29°C). The entire population died after several generations due to the loss of complementing mt:CoI activity from *mt:ATP6*[1]. However, spontaneous recombination in some lineages

generated mitochondrial genomes with the functional *mt:CoI* from *mt:ATP6*[1] and the selfish drive from the ts genome, which allows these lineages to survive and propagate at 29°C (Ma and O'Farrell, 2015; Ma and O'Farrell, 2016; Fig. 4 A). Here, we re-

**Figure 4. REC mediates spontaneous recombination and safeguards mtDNA during ageing. (A)** Numbers of recombinant mitochondrial genomes generated in *REC-Halo* and *REC^{ΔMTS-Halo}* flies heteroplasmic for *mt:ATP6[1]* and *mt:ND2^{Δ1}+CoI^{T300I}* from 108 lineages. **(B)** Stacked bar plots showing the percentage of flies with different levels of mtDNA mutations. The number of mtDNA variants (above 1% heteroplasmy) detected in individual flies was divided by the average number of mtDNA variants detected in 2-d-old *REC-Halo* fly population to calculate the fold change in mtDNA mutation levels (*n* = 6–8 flies for each genotype of a given age). **(C)** Relative ATP levels and climbing distance in young and aged flies (ATP levels: *n* = 6–10 groups of 5 males; average distance climbed: *n* = 9–14 groups of 8 males). Data represent mean ± SD, two-way ANOVA with Dunnett's multiple comparisons, ns P > 0.05, *P < 0.05, **P < 0.01, ****P < 0.0001.

generated this heteroplasmic line and measured the frequency of isolating recombinant mtDNA by sequencing the survived lines at 29°C. As *REC^{KO}-Halo* flies showed reduced female fertility at 29°C (Fig. S3 A), we only compared the *REC^{ΔMTS}* flies with controls. We found that reducing mitochondrial REC is sufficient to diminish spontaneous recombination: out of the 108 heteroplasmic lineages we followed, 13 and 2 lineages with recombinant mtDNA were isolated for *REC-Halo* and *REC^{ΔMTS}-Halo* flies, respectively (Fig. 4 A and Fig. S3 B). Therefore, REC is also important for mtDNA recombination under physiological conditions.

To investigate the role of REC in safeguarding mtDNA during normal development and ageing, we sequenced mtDNA of young and old flies to detect mtDNA variants above 1% heteroplasmy. No novel insertions or deletions were detected in any samples. The number of single nucleotide polymorphisms (SNPs) in young flies was slightly higher for *rec* mutants. This was accentuated in aged flies, with a higher proportion of individuals carrying more SNPs than controls (Fig. 4 B). The increased mtDNA mutation load in aged *rec* mutants was associated with reduced mitochondrial function and healthspan, as shown by reduced ATP levels and locomotion in aged *REC^{ΔMTS}* and *REC^{KO}* flies (Fig. 4 C). There was no reduction in mtDNA copy number (Fig. 2 A) or expression of nuclear and mitochondrial genes encoding mitochondrial proteins in either young or old *rec* mutant flies (Fig. S3 C). We thus conclude that *rec* mutants

accumulate more mtDNA mutations during ageing, and this could contribute to reduced mitochondrial function and healthspan, irrespective of mitochondrial gene expression and mtDNA copy number.

REC is evolutionarily diverged from its human ortholog MCM8, and *Drosophila* lack an ortholog of MCM9 (Kohl et al., 2012). By immunostaining, we found that a large proportion of MCM8 co-localized with mitochondria and mtDNA in HeLa cells (Fig. 5 A and Fig. S4 A). Blotting subcellular fractions using a different MCM8 antibody also showed that MCM8 is predominantly in the mitochondrial fraction, although it was also found in cytoplasmic and nuclear fractions (Fig. 5 B). Like *Drosophila* REC and other mitochondrial matrix proteins TFAM and PolG1, MCM8 in the crude mitochondrial fraction was resistant to proteinase K digestion after the mitochondrial outer membrane was permeabilized (Fig. 5 C). Furthermore, we performed co-immunoprecipitation with both TFAM and MCM8 antibodies and found that MCM8 co-immunoprecipitated with TFAM (Fig. 5 D and Fig. S4 B). Together, these data suggest that MCM8 localizes proximal to mtDNA nucleoids in the matrix.

To examine the role of MCM8 in mitochondria and mtDNA maintenance, we generated an MCM8 knockout line by CRISPR/Cas9-based editing (Fig. 5 E). MCM8$^{KO}$ cells have similar growth rates, mtDNA copy numbers and ATP levels as controls when cultured in standard media with glucose (Fig. 5 F and Fig. S4 C). However, they carry 4.6 times as many mtDNA mutations as control cells after 22 generations of passage (Fig. 5 G). When cultured in galactose media, MCM8$^{KO}$ cells show much-reduced ATP levels, whereas the total mtDNA copy number remains unchanged (Fig. 5 F). Their mitochondria also have lower membrane potential after growing in galactose media for 2 d (Fig. 5 F). We conclude that, like REC, MCM8 safeguards mtDNA to maintain mtDNA integrity and mitochondrial function.

This study provides clear evidence that REC is required for mtDNA recombination and DSB repair in vivo. Crucially, compromising REC or MCM8 is sufficient to drive mitochondrial mutation accumulation and reduce the healthspan. Identifying the detailed mechanism of mtDNA recombination is challenging as the enzymatic role of REC and MCM8 has not yet been defined. In *Drosophila*, REC is suggested to function downstream of the RAD51 recombinase to facilitate DNA repair synthesis and generate meiotic crossovers (Blanton et al., 2005). In human cells, MCM8 performs a similar role and is recruited to sites of DNA damage by HROB (Hustedt et al., 2019), but can also function earlier in mitotic recombination by recruiting the MRE11-RAD50-NBS1 (MRN) complex (Lee et al., 2015) and RAD51 (Park et al., 2013) to promote DSB resection and HR repair. REC-mediated mtDNA recombination could operate in a similar manner as recombination in the nucleus. Previous studies have reported that RAD51 and some of its paralogs are mitochondrial in human cell lines by performing subcellular fractionation and immunofluorescence (Dahal et al., 2018; Mishra et al., 2018; Sage et al., 2010). Similarly, MRE11 has been reported to function in mitochondria of human and mouse cell lines (Dahal et al., 2018; Dmitrieva et al., 2011; Li et al., 2019; Luzwick et al., 2021). However, the RAD51 homologues and MRN complex

proteins that we tested showed no mitochondrial enrichment in *Drosophila* oocytes (Fig. S1 A). Alternatively, REC-mediated mtDNA recombination could resemble that of the T7 bacteriophage, given that the mtDNA replication machinery resembles the T7 phage replisome (Falkenberg and Gustafsson, 2020). The detailed mechanism of T7 phage recombination remains unclear, but it is known to be replication-based and relies on a single-strand DNA binding (SSB) protein and a helicase to mediate homologous strand exchange (Kong and Richardson, 1996; Yu and Masker, 2001). Accurate T7 recombination can occur with only 8 bp homology (Lin et al., 2012), which we also observed for mtDNA recombination in the *Drosophila* germline (Fig. S2 D; Ma and O'Farrell, 2015). REC could be the helicase that works with mtSSB and PolG1 to mediate a T7-phage like recombination in animal mitochondria.

Notably, the loss of REC does not completely abolish mtDNA recombination (Fig. 3, C and D). Similarly, meiotic crossover events are still detected in *rec* null mutants, although the frequency is reduced to ~10% of the wild-type level (Blanton et al., 2005). It is likely that other components or pathways exist inside mitochondria to facilitate the remaining recombination events in the absence of REC. Alternatively, non-crossover gene conversion, which has been shown to be independent of REC in meiotic recombination (Blanton et al., 2005), could operate at low frequencies to achieve precise DSB repair in somatic and germline mitochondria.

Irrespective of how recombination is achieved, this study demonstrates the importance of recombination in safeguarding animal mtDNA during evolution and ageing. Although spontaneous recombination in animal mitochondria is rare, occasional recombination in the germline could be sufficient to prevent the accumulation of deleterious mutations that may otherwise occur by Muller's ratchet in asexual populations (Charlesworth et al., 1993; Neiman and Taylor, 2009). REC-mediated repair can also reduce pathogenic mitochondrial mutations to further aid the conservation of mtDNA in individual maternal lineages. In the soma, REC-mediated mtDNA repair may limit the progression of ageing and mitochondrial diseases (Fig. 3 D and Fig. 5 G). Cancers and premature ovarian failure are also associated with mtDNA mutations (Gammage and Frezza, 2019; Tiosano et al., 2019) and present in patients with MCM8/9 mutations (AlAsiri et al., 2015; Cai et al., 2015; He et al., 2017; Heddar et al., 2020; Morii et al., 2019; Tenenbaum-Rakover et al., 2015; Wang et al., 2020). It is possible that mtDNA and nuclear genome instabilities couple to cause disease in patients with MCM8/9 mutations. Future investigation is needed to better understand how mtDNA maintenance proteins contribute to ageing and disease.

## Materials and methods

### *Drosophila* husbandry and stocks

All fly stocks were raised on standard media at 25°C unless otherwise stated. Lines used in this study include *nos-Cas9* (BDSC: 54591), *nubbin-GAL4* (BDSC:42699), *nos-GAL4* (BDSC:64308), *Ubi-mtSSB-GFP* (a gift from Patrick O'Farrell, University of California, San Francisco), Opa1-3HA (Liu et al., 2020), *mt:ND2$^{del1}$+CoI$^{T300I}$*

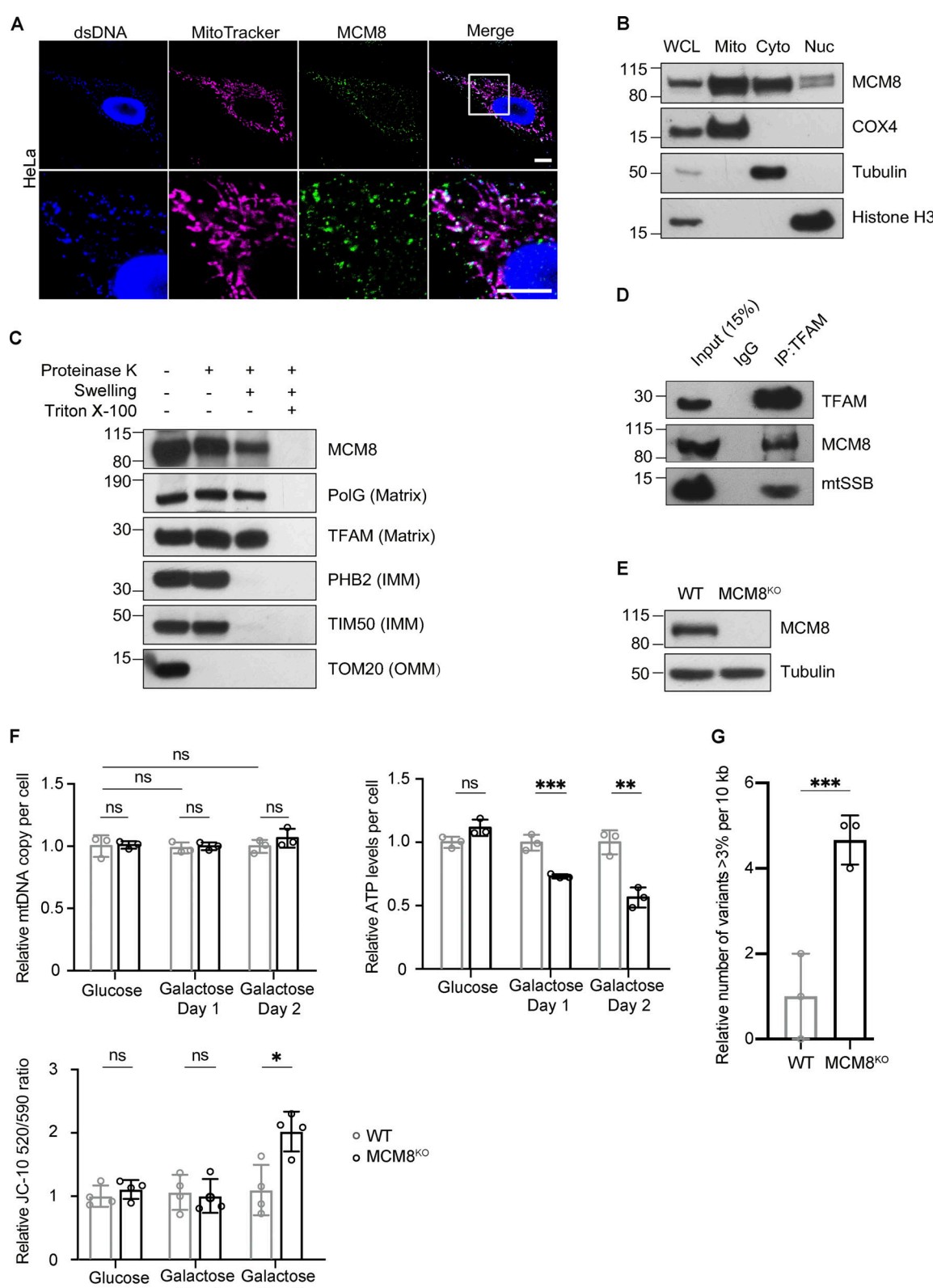

Figure 5. **MCM8 co-localizes with mitochondria and mtDNA to safeguard mtDNA in human cells. (A)** Representative images of HeLa cells with MitoTracker-labeled mitochondria and immune-stained with antibodies against double-stranded DNA (dsDNA) showing co-localization of MCM8 with mitochondria and mtDNA. Scale bar: 10 μm. **(B)** Immunoblot of whole-cell lysate (WCL), mitochondrial (Mito, anti-COX4), cytoplasmic (Cyto, anti-tubulin), and nuclear (Nuc, anti-Histone H3) fractions of HEK293T cells to test the subcellar enrichment of MCM8. **(C)** Mitochondrial fractions post proteinase K treatments immunoblotted with anti-MCM8, anti-PolG1, anti-TFAM, anti-prohibitin 2 (PHB2), anti-TIM50, and anti-TOM20 antibodies, suggest that MCM8 localizes to the mitochondrial matrix. **(D)** MCM8 co-immunoprecipitated with TFAM in HEK293T cells. Anti-TFAM antibody was used to pull down MCM8 and mtSSB. Co-

immunoprecipitation with anti-MCM8 antibodies also pulled down TFAM (Fig. S4 B). **(E)** Immunoblot confirms the absence of MCM8 protein in knockout cells (Fig. S4 D). Tubulin was blotted as the loading control. The MCM8$^{KO}$ cells carry a 2 bp deletion that results in an early premature stop codon. **(F)** qPCR and ATP assays, and JC-10 staining measuring mtDNA copy number, ATP levels and mitochondrial membrane potential of MCM8$^{KO}$ and wild-type cells cultured in glucose or galactose media ($n$ = 3 biological replicates). Data represent mean ± SD, Student's one-sided $t$ test; ns, $P > 0.05$; *, $P < 0.05$; **, $P < 0.01$; ***, $P < 0.005$. **(G)** The number of mtDNA variant sites was increased in MCM8$^{KO}$ cells. Only the coding region with consistently high coverage was used for analysis and only variants present in >3% of reads were counted to avoid false positives associated with read strand biases observed with those present in <3% ($n$ = 3 biological replicates). Data represent mean ± SD, Student's one-sided $t$ test; ***, $P < 0.005$. Source data are available for this figure: SourceData F5.

(Ma et al., 2014), *D. melanogaster* (*mt:mauritiana*; Ma and O'Farrell, 2016), *w1118* and *w*;attP40*. *REC-Halo*, *REC-FLAG*, *REC$^{ΔMTS}$-Halo*, *REC$^{KO}$-Halo*, *HDM-Halo*, *PolG1-Halo*, *XLF1(CG12728)-Halo*, *SPN-B-Halo*, *SPN-D-Halo*, *MRE11-Halo*, *NBS-Halo*, *UASp-REC-GFP*, *UASp-mito-NciI*, and *mt:NciI$^{resistant}$* were generated for this paper.

### Cell lines and maintenance
*Drosophila* S2R+ cells were maintained in Schneider's medium (21720024; Thermo Fisher Scientific) with 10% fetal bovine serum (10500064; Thermo Fisher Scientific) and 1% penicillin-strepto-mycin-glutamine (10378016; Thermo Fisher Scientific) at 25°C.

HeLa, Hela Cas9, HEK293T, and MCM$^{KO}$ cells were maintained in complete DMEM media (31966047; Thermo Fisher Scientific) supplemented with 10% fetal bovine serum and 1% penicillin-streptomycin (15070063; Thermo Fisher Scientific) at 37°C. To test growth under different carbon sources, cells were maintained in DMEM supplemented with 10% FBS, 1% penicillin-streptomycin, GlutaMAX (35050038; Thermo Fisher Scientific), and either 25 mM D-glucose (A2494001; Thermo Fisher Scientific) or 10 mM D-galactose (G5388; Merck).

### Establishing *Drosophila* lines with certain nuclear and mitochondrial genotypes
Mutant generation and endogenous tagging were performed by CRISPR-Cas9-based editing as described in FlyCRISPR (https://flycrispr.org/). gRNAs were designed using FlyCRISPR target finder (http://tools.flycrispr.molbio.wisc.edu/targetFinder) and cloned into a pCFD5 plasmid. Donor plasmids were generated using InFusion Cloning (638911; Takara Bio). Plasmids were injected into *nos-Cas9* or *nos-Cas9;;REC-Halo* embryos to establish individual stocks. The nature of mutations and tagging was verified by Sanger sequencing. To make transgenic lines with UASp for GAL4-driven expression, open reading frames of candidate genes were cloned into the pPWG or pPW vector. Plasmids were injected into y[1] w*;attP40 *or w1118* flies to establish individual stocks. All primers used for this section are listed in Table S1.

Flies heteroplasmic for *mt:mel* and *mt:mau* were generated by cytoplasmic transplantation (Ma et al., 2014). Flies heteroplasmic for *mt:mel* and *mt:yak* were established in a previous study (Ma and O'Farrell, 2016). Flies homoplasmic for *mt:NciI$^{resistant}$* were isolated by germline expression of mitochondrially targeted NciI driven by *nos-GAL4* (Xu et al., 2008). Individual lines were established from resulting progeny that were homoplasmic for a mitochondrial genotype carrying a mutation in the NciI recognition site. The mtDNA genotype was confirmed by PCR and by Sanger sequencing using primers listed in Table S1.

### S2R+ cell co-localization screen
Open reading frames of candidate genes known or predicted to have a direct role in nuclear DNA repair were cloned into pENTR3C by InFusion Cloning (see Table S1 for all relevant primers). They were transferred from pENTR3C plasmids into the pAWG destination vector by Gateway Cloning (11791100; Thermo Fisher Scientific). S2R+ cells were transiently transfected using TransIT-Insect Transfection Reagent (MIR6100; Mirus Bio). One day after transfection, cells were incubated with 2 µM Hoechst (62249; Thermo Fisher Scientific) and 0.1 µM MitoTracker Red FM (8778; Cell Signaling Technology) and imaged on an SP5 Leica inverted confocal microscope.

### Preparation of fly extracts and subcellular fractionation
Dissected ovaries were resuspended in PBS in parallel preparations. The harvested tissues were centrifuged at 17,000 × $g$ for 5 min at 4°C, washed with PBS, and homogenized in Pierce IP lysis buffer (87787; Thermo Fisher Scientific) containing 1% EDTA-free protease inhibitor cocktail (11836170001; Merck). The homogenate was centrifuged at 17,000 × $g$ for 10 min at 4°C to separate soluble proteins from debris. Soluble protein concentrations were determined using the Pierce BCA protein assay kit (23225; Thermo Fisher Scientific).

To prepare whole extracts and subcellular fractions from embryos, overnight embryos were collected, dechorionated using 50% bleach and homogenized in mitochondrial isolation buffer (250 mM sucrose, 10 mM Tris, 10 mM EDTA, 1% BSA, pH 7.4, 1% EDTA-free protease inhibitor cocktail). Homogenate was centrifuged at 100 × $g$ for 5 min at 4°C, and the resulting supernatant was collected as the whole-cell lysate. To obtain the mitochondria fraction, the whole-cell lysate was centrifuged at 1,000 × $g$ for 5 min at 4°C twice, and the resulting supernatant was centrifuged at 10,000 × $g$ for 5 min at 4°C. The pellet was washed with mitochondrial suspension buffer (250 mM sucrose, 10 mM Tris, 10 mM EDTA, 0.15 mM MgCl$_2$, pH 7.4, 1% EDTA-free protease inhibitor cocktail). The final mitochondrial pellet was resuspended in a mitochondrial suspension buffer. To separate cytosolic and nuclear fractions, the whole-cell lysate was centrifuged at 1,000 × $g$ for 5 min at 4°C. The pellet was washed twice and solubilized with mitochondrial suspension buffer + 1% NP-40 to yield the nuclear fraction. The supernatant was centrifuged at 1,000 × $g$ for 5 min at 4°C, then at 10,000 × $g$ for 5 min at 4°C, and the resulting supernatant was supplemented with 1% NP-40 to yield the cytosolic fraction. Soluble protein concentrations were determined using the Pierce BCA protein assay kit.

### Preparation of human cell extracts and subcellular fractionation
Trypsinized cells were washed once with 4°C PBS and incubated with Pierce IP lysis buffer containing 1% EDTA-free protease

inhibitor cocktail for 30 min at 4°C. Whole-cell lysates were centrifuged at 17,000 × *g* for 10 min at 4°C to separate soluble proteins from cell debris. Subcellular fractions were prepared from cells using the cell fractionation kit (ab109719; Abcam) in the presence of 1% EDTA-free protease inhibitor cocktail. Soluble protein concentrations were determined using the Pierce BCA protein assay kit.

## Western blot analysis

Whole-cell lysates, subcellular fractions, and immunoprecipitates were supplemented with 100 mM Bolt sample reducing agent (B0009; Thermo Fisher Scientific), incubated for 5 min at 95°C, and separated on a Bolt 4–12%, Bis-Tris gel (NW04120BOX; Thermo Fisher Scientific) in MES SDS running buffer (B0002; Thermo Fisher Scientific). They were transferred to Immobilon-P PVDF membranes (IPVH00010; Merck) in a Tris/glycine transfer buffer (25 mM Tris, 192 mM glycine, 20% methanol, 0.1% SDS, pH 8.3). The blots were blocked with 10% milk for 1 h at room temperature and probed for FLAG (mouse, F1804, 1:500; Merck), Halo (mouse, G9211; Promega), MCM8 (mouse, 16541-1-AP; Thermo Fisher Scientific; rabbit, PA5-65399; Thermo Fisher Scientific; mouse, H00084515-M02; Novus Biologicals), mtSSB (rabbit, 12212-1-AP; Proteintech) or TFAM (mouse, ab119684; Abcam). Antibodies against COX4 (mouse, ab33985; Abcam), tubulin (rat, ab6160; Abcam), GAPDH (mouse, 60004-1; Proteintech), and histone H3 (rabbit, ab1791; Abcam) or PCNA (mouse, ab29; Abcam) were used to serve as mitochondrial, cytosolic, and nuclear markers, respectively. Western blots were visualized by film exposure using horseradish peroxidase-conjugated secondary antibodies against chicken (A16054; Thermo Fisher Scientific), mouse (62-6520; Thermo Fisher Scientific), rabbit (HAF008; Novus Biologicals), or rat (18-4818-82; Thermo Fisher Scientific), in combination with Clarity Western ECL substrate (1705061; Bio-Rad Laboratories) on an SRX-101A developer.

## Co-immunoprecipitation

Trypsinized human cells were washed three times with PBS, and homogenized with a Dounce homogenizer together with IP lysis buffer (50 mM Tris-HCl, pH 7.4, 100 mM NaCl, 1 mM EDTA, 5% glycerol, 0.1% SDS and 1% NP-40). The lysates were incubated on ice for 20 min and centrifuged at 12,000 × *g* at 4°C for 10 min to remove cell debris. 2–4 mg of cell extracts were incubated overnight at 4°C with 2 μg of mouse IgG (A0919; Merck), TFAM (mouse, ab119684; Abcam), or MCM8 (mouse H00084515-M02; Novus Biologicals) antibody. Immunoprecipitates were collected with high-affinity protein G agarose beads (ab193258; Abcam) for 3 h at 4°C, centrifuged at 200 *g* at 4°C for 5 min, and washed five times with IP lysis buffer. Samples were eluted with Bolt LDS sample buffer for 5 min at 95°C, and separated on a Bolt 4–12% gel as described above.

## Proteinase K protection assay

The proteinase protection assay was performed as previously described (Le Vasseur et al., 2021). Briefly, HEK293-T cells or *Drosophila* ovarian tissues were homogenized in homogenization buffer (210 mM mannitol, 70 mM sucrose, 10 mM HEPES, 1 mM

EDTA, pH 7.4) plus 1% protease inhibitor cocktail (11836170001; Merck). The cell debris and nucleus were removed by centrifuging at 500 × *g* for 5 min at 4°C and then 1,000 × *g* for 5 min at 4°C. The mitochondria were pelleted by centrifuging at 5,000 × *g* for 10 min at 4°C and washed at 5,000 × *g* for 10 min at 4°C in homogenization buffers without protease inhibitors. The mitochondrial pellet was then suspended in homogenization buffers and the protein concentration was measured using Qubit protein assay kit (Q33211; Thermal Fisher Scientific). For the proteinase K protection assay, 50 mg of mitochondrial proteins were pelleted by centrifugation at 5,000 × *g* for 10 min at 4°C and re-suspended in 500 μl of homogenization buffer, mitoplast/swelling buffer (10 mM HEPES, pH 7.4), or solubilizing buffer (homogenization buffer with 1% Triton X-100), and incubated on ice for 15 min. The mitoplast/swelling sample was pipetted up and down 15 times to disrupt the mitochondrial outer membrane. Proteinase K (P8107S; New England Biolabs) was then added to the samples to a final concentration of 4 U/ml, and samples were incubated on ice for 20 min. To terminate the reaction, PMSF was added to all samples to a final concentration of 2 mM followed by 5 min incubation on ice. The resulting proteins were then precipitated by 12.5% TCA, washed with cold acetone, re-suspended in 100 μl of 1× LDS sample buffer, and boiled for 5 min. Finally, 20 μl of samples were analyzed by Western blot. Tom 20 (rabbit, ab186735; Abcam), TIM50 (rabbit, 22229-1-AP; Proteintech), PHB2 (mouse, 66424-1-1g; Proteintech), TFAM (mouse, ab119684; Abcam), and PolG1 (rabbit, ab128899; Abcam) were probed as markers for different compartments of human mitochondria. Porin (rabbit, PC546; Merck), Opa1-HA (HA antibody: rabbit, 3724S; Cell Signaling Technology), PolG1-Halo (Halo antibody: mouse, G9211; Promega), and ATP5A (mouse, ab14748; Abcam) were probed as markers for different compartments of *Drosophila* mitochondria.

## RNA extraction and RT-qPCR

Total RNA from whole flies, dissected tissues (8 ovaries, 5 male heads, or 16 larval imaginal discs per sample), or human cells was extracted using TRIzol (15596026; Thermo Fisher Scientific). Extracted RNA was further purified using the Qiagen RNeasy kit with on-column DNase treatment (74004; Qiagen). Purified RNA was then reverse-transcribed using RevertAid First-strand cDNA synthesis kit (K1621; Thermo Fisher Scientific) and Oligo $(dT)_{18}$ primers. The relative mRNA level of individual genes was measured by qPCR using 2× SensiFast SYBR Green PCR Master Mix (98020; Bioline) and normalized to *ef1a*. All the primers used are listed in Table S1.

## Immunofluorescence and confocal imaging

Dissected ovaries were fixed in 4% paraformaldehyde (pH 7.4) for 30 min, washed three times in PBS with 0.1% TritonX-100 (0.1% PBST), and incubated in 0.5% PBST overnight at 4°C. They were then washed once with PBS followed by Halo staining in 0.5 μM HaloTag TMR ligand (G8251; Promega) for 1 h at room temperature. To minimize background signal, ovaries were further washed three times in 0.1% PBST and incubated overnight at 4°C. If subsequent immunostaining to visualize mitochondria was required, ovaries were incubated with ATP5A

antibodies (mouse, ab14748; Abcam) in 0.1% PBST supplemented with 3% BSA overnight at 4°C, followed by another overnight incubation with Alexa Fluor 647 (A21235; Invitrogen) in 0.1% PBST with 3% BSA. Ovaries were then mounted in Vectashield medium with DAPI (H-1200; Vector Laboratories) and imaged on a Leica SP8 DM6000 CS upright confocal microscope using a 63× oil lens 1.4NA and Leica LASX Acquisition software at room temperature. Images were processed using ImageJ and Affinity Designer software.

For mammalian cells, $10^5$ cells were seeded into each well of a 24-well glass-bottomed plate (82406; Thistle Scientific). The next day, cells were incubated with 0.1 μM MitoTracker Red FM for 30 min, washed three times with PBS, and fixed by 4% paraformaldehyde (pH 7.4) for 10 min at 37°C. The cells were washed three times with PBS and then permeabilized by incubation with 0.1% PBST for 15 min at room temperature. Following another three washes with PBS, cells were incubated with 2% BSA in PBS for 4 h at room temperature. Primary antibodies were then added in PBS with 0.1% BSA and incubation continued overnight at 4°C. Primary antibodies used include dsDNA (mouse, ab27156; Abcam) and MCM8 (rabbit, PA5-65399; Thermo Fisher Scientific; rabbit, ab191914; Abcam). Following incubation with primary antibodies, the cells were washed three times with PBS and incubated with secondary Alexa Fluor 546 (mouse, A11030; Thermo Fisher Scientific) or Alexa Fluor 488 (rabbit, A21206; Thermo Fisher Scientific) antibodies in PBS with 0.1% BSA for 45 min at room temperature. The cells were washed three times with 0.1% PBST and then incubated with 2 μM Hoechst and imaged on an SP5 Leica inverted confocal microscope using a 63× oil lens 1.4 NA and Leica LASX Acquisition software at room temperature. Images were processed using ImageJ and Affinity Designer software.

For experiments with drug treatments prior to fixation, ovaries were dissected from flies expressing mtSSB-GFP and incubated for 2 h in PBS with 50 μg/ml bleomycin (A8331; ApexBio), 1 mM paraquat (856177; Sigma-Aldrich), 10 mM $H_2O_2$ (VWR, 23615.261), 50 μM cisplatin (P4394; Sigma-Aldrich), or PBS alone, followed by fixation and Halo staining as described above. All drugs were dissolved in PBS. Mitochondrial areas were defined by mtSSB-GFP fluorescence, and the average Halo signals in the defined mitochondrial areas were quantified using ImageJ. Egg chamber mitochondria were selected as regions of interest by thresholding of the mtSSB-GFP, and then the mean pixel intensity (AU) of the Halo channel was measured and normalized to the controls.

To measure mitochondrial membrane potential, 50,000 cells were seeded in a well of 18-well Ibidi slide (81816; Thistle Scientific) and incubated with 7.5 mM JC-10 (22204; AAT Bio) for 30 min in media before imaging with an SP5 Leica inverted confocal microscope at Ex/Em = 490/525 nm and 540/590 nm, using a 63× oil lens 1.4NA and Leica LASX Acquisition software at room temperature, followed by ratio analysis using ImageJ.

### Recombination assay
*UAS-mito-NciI;REC-Halo*, *UAS-mito-NciI;REC$^{ΔMTS}$-Halo*, and *UAS-mito-NciI; REC$^{KO}$-Halo* fly lines were generated by genetic crosses to carry the following mtDNA genotypes: *mt:mau/mt:*

*mel* heteroplasmy, *mt:yak/mt:mel* heteroplasmy, *mt:ATP[1]/mt: ND2$^{ΔI}$+CoI$^{T300I}$* heteroplasmy, or *mt:NciI$^{resistant}$* homoplasmy. *REC\*-Halo* is used to represent *REC-Halo*, *REC$^{ΔMTS}$-Halo*, or *REC$^{KO}$-Halo* flies below. For the germline assay, heteroplasmic *UAS-mito-NciI;REC\*-Halo* females were crossed to *nos-GAL4; REC\*-Halo* males (Fig. S2 C). Individual F1 females were then mated to *nos-GAL4* males to generate F2 progeny. Adult progeny from F1 females were genotyped for their mtDNA by Sanger sequencing using primers listed in Table S1. For the somatic assay, heteroplasmic *UAS-mito-NciI;REC\*-Halo* females were crossed to *nubbin-GAL4/CyO* males. The percentage of rescue was calculated by dividing the number of progeny without *CyO* by the number of progeny with *CyO* (Fig. S2 F).

### mtDNA copy number measurement
The total copy number of mtDNA in newly laid eggs was measured as described in Chiang et al. (2019). In brief, for each genotype, 20 eggs collected within 20 min of laying were lysed in 100 μl of QuickExtract buffer (QE09050; Lucigen) using a BeadBug microtube homogenizer (Z763713; Sigma-Aldrich) and prefilled tubes (Z763799; Sigma-Aldrich). Samples were agitated twice at 4,000 rpm for 60 s and then incubated for 15 min at 65°C and 5 min at 95°C. The total mtDNA copy number was then measured by qPCR using *mt:CoI* primers and normalized to the number of eggs (Table S1). For fly somatic tissues and human cells, qPCR was performed with both mitochondrial and nuclear DNA primers (Table S1), and the relative mtDNA copy number per cell was calculated based on the Ct value differences.

### Phenotypic assays
Prior to phenotypic assays, males of *REC-Halo*, *REC$^{ΔMTS}$-Halo*, or *REC$^{KO}$-Halo* were crossed to *w1118* females with balancer chromosomes to establish *REC-Halo*, *REC$^{ΔMTS}$-Halo*, or *REC$^{KO}$-Halo* fly lines with an isogenic nuclear (except for the *rec* locus) and mitochondrial background (i.e., *w1118* mtDNA). For the fertility assay, newly eclosed females of the same genotype were placed in a vial with *w1118* or *nos-GAL4* males. *w1118* or *nos-GAL4* males were used instead of *REC\*-Halo* flies to minimize the impact of male fertility on female egg laying and hatching. The flies were flipped into a new vial each day or every 2–3 d, and the number of eggs laid and adult progeny produced in each vial were counted for a certain number of days and normalized. For the climbing assay, eight adult males of a specified age were placed in a sealed measuring cylinder, which was tapped several times to knock all the flies to the bottom. The distance climbed upwards by each fly in 30 s was recorded and the average distance climbed was calculated. For the longevity assay, 20 flies were placed in each vial on the day of their eclosion. The vials were flipped every 2 d, and survival was recorded.

### ATP measurements
ATP levels were measured using the ATP determination kit (A22066; Thermo Fisher Scientific). Five adult males of a specified age were homogenized in 100 μl of 6 M guanidine HCl (24115; Thermo Fisher Scientific) and 0.01 M Tris-HCl (pH 7.3) and frozen in liquid nitrogen for 5 min. The sample was then incubated at 95°C for 5 min and centrifuged at 12,000 × *g* for

10 min at 4°C. 10 μl of the sample was used for each 100 μl reaction as instructed in the kit manual. ATP levels for each sample were normalized to protein concentration measured using the Pierce BCA protein assay kit.

### Generation of mutant human cell lines
The MCM8 mutant cell line was generated using a HeLa Cas9 line. gRNAs were designed using the IDT design tool for Alt-R CRISPR-Cas9 guide RNA. crRNA and tracrRNA were synthesized by IDT and mixed in a 1:1 crRNA:tracrRNA 30 μM duplex solution, incubated at 95°C for 5 min, cooled and diluted to 1 μM. Lipofectamine RNAiMax (13778075; Thermo Fisher Scientific) was used to transfect duplexed gRNA into HeLa Cas9 cells. Single-cell clones were isolated, expanded, and genotyped by Sanger sequencing. Primers and crRNA (gRNA) sequences are listed in Table S1.

### Sequencing and read analysis
*REC-Halo*, *REC*$^{\Delta MTS}$*-Halo*, and *REC*$^{KO}$*-Halo* fly lines in an isogenic nuclear (except for the *rec* locus) and mtDNA background were maintained in parallel before sequencing. Total DNA of individual flies was extracted as described in Ma et al. (2014). In brief, individual adults were squashed in 100 μl of homogenization buffer (100 mM Tris-HCl [pH 8.8], 10 mM EDTA, 1% SDS) and incubated at 70°C for 30 min. Potassium acetate was added (to a final concentration of 1 M), and samples were incubated on ice for 30 min. Samples were centrifuged at 20,000 *g* for 10 min at room temperature. DNA was recovered from the supernatant by adding 0.5× volume of isopropanol, followed by washing with 70% ethanol. DNA was then dissolved in 10 μl H$_2$O before further dilution. Total DNA of HeLa cell lines was extracted using DNeasy Blood & Tissue Kit (69504; Qiagen). DNA was prepared for sequencing using the Nextera XT DNA Library preparation kit (FC-131-1096; Illumina) and index kit (FC-131-2001; Illumina) following manufacturer's instructions. Paired ends were sequenced using Illumina NovaSeq6000. Reads were aligned against the *Drosophila* genome (BDGP Release 6 + ISO1 MT/dm6) or human genome (hg38). Integrated Genome Viewer was used to identify mtDNA sites with variants that were above 1% read threshold for fly samples. The threshold was determined to ensure the independence of the average read coverage and number of variants. For HeLa cells, we used 3% read threshold because variants below 3% show extreme/strong strand biases, indicating a potential high false-positive rate for these SNPs. Due to the variation in genome coverage observed, mtDNA coding regions with consistently high coverage were used for analysis: 1,300–14,730 bp for fly samples; and 594–3,511, 3,629-13,631, and 13,870–16,560 bp for HeLa cells.

### Statistical analysis
Statistical analyses were performed using GraphPad 9 software. The Student's *t* test or one-way ANOVA test was used to analyze the effect of a single variable, and the two-way test was used to analyze the effect of more than one variable. Dunnett's multiple comparisons test was used after ANOVA tests to report significance. For *t* tests, data distribution was assumed to be normal, but this was not formally tested.

Significance was defined by *P < 0.05, **P < 0.01, and ***P < 0.005.

### Online supplemental material
Online supplemental material includes the visualization of endogenous REC and other repair factors in *Drosophila* ovaries (Fig. S1), additional data supporting REC's role in mediating mtDNA homology-based repair (Fig. S2) and spontaneous recombination (Fig. S3), as well as further evidence of mitochondrial localization of MCM8 and the characterization of the knockout cell line (Fig. S4). Table S1 lists all the candidate genes examined for the mitochondrial enrichment of their protein products when overexpressed in *Drosophila* S2R+ cells (Fig. 1 A), and primers and gRNA used in this study.

### Data availability
All data are available in the main text or supplementary materials. mtDNA sequencing data are deposited to Genbank: BioProject accession no. PRJNA796012, and all fly stocks and cell lines will be made available to the scientific community upon request.

## Acknowledgments
We thank Professor Steve Jackson and Dr. Francisco Muñoz Martinez (Cancer Research UK Cambridge Centre, UK) for sharing the HeLa and HeLa Cas9 cell lines, and Dr. Emma Rawlins (Gurdon Institute, University of Cambridge, UK) for sharing the HEK293T cell line. We thank the Gurdon Institute Imaging and other Core Facilities for their support. We thank Professor Julie Ahringer (Gurdon Institute, University of Cambridge, UK) for critical reading of this manuscript.

This work is funded by Wellcome Trust Studentship 203767/Z/16/Z to A. Klucnika, Chinese Council Scholarship 202008440165 to P. Mu, Wellcome Sir Henry Dale Fellowship 202269/Z/16/Z and ERC Starting Grant 803852 to H. Ma, and Wellcome Trust grant 203144 and CRUK grant C6946/A24843 for the Gurdon Institute Core Facility.

The authors declare no competing financial interests.

Author contributions: Conceptualization: A. Klucnika, H. Ma; Methodology: A. Klucnika, P. Mu, H. Ma; Investigation: A. Klucnika, P. Mu, J. Jezek, M. McCormack, Y. Di, C.R. Bradshaw, H. Ma; Visualization: A. Klucnika, P. Mu; Funding acquisition: A. Klucnika, P. Mu, H. Ma; Writing—original draft: A. Klucnika, H. Ma; Writing—review & editing: A. Klucnika, P. Mu, J. Jezek, M. McCormack, Y. Di, C.R. Bradshaw, H. Ma.

Submitted: 26 January 2022

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

# Supplemental material

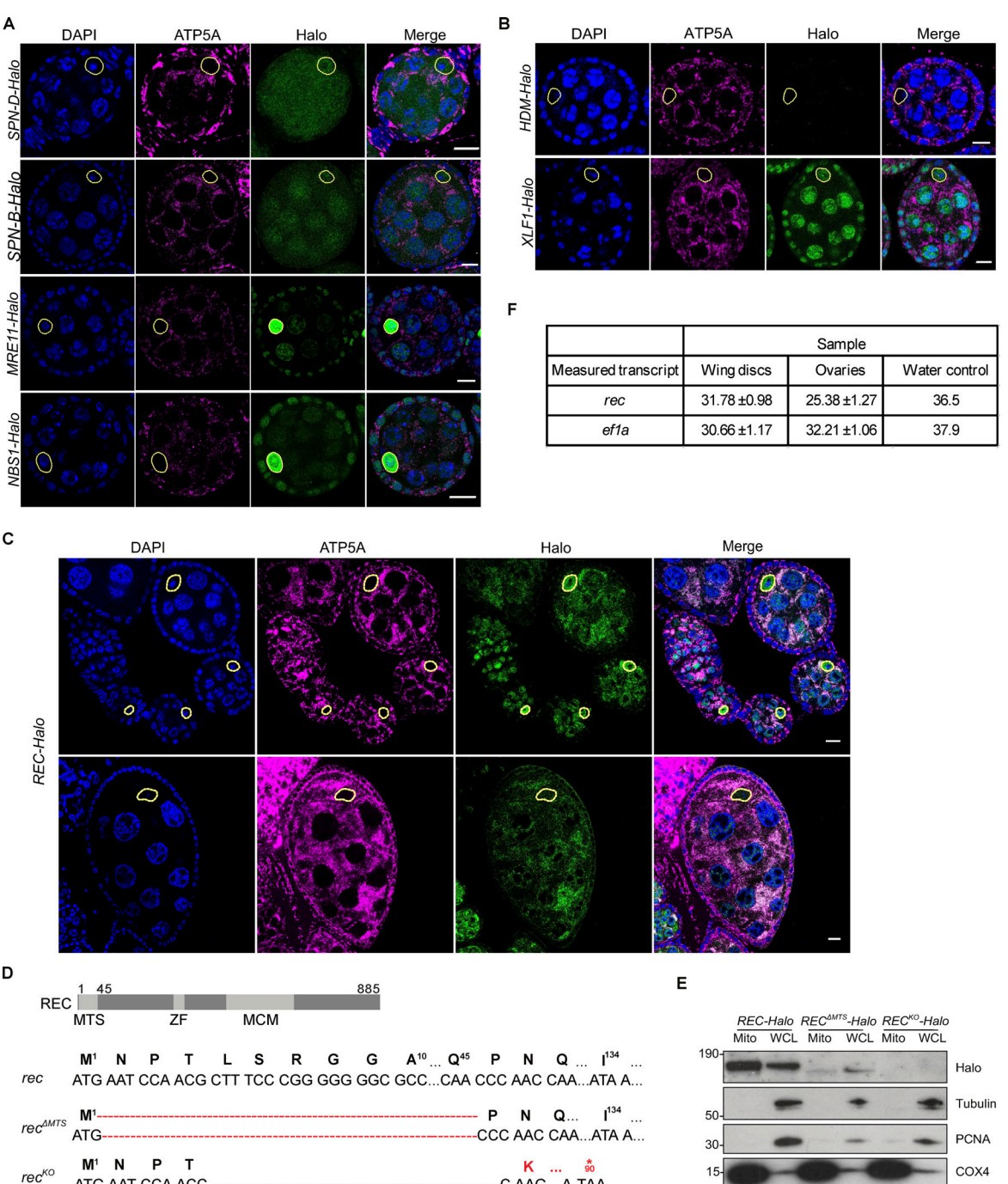

| | | Sample | |
|---|---|---|---|
| Measured transcript | Wing discs | Ovaries | Water control |
| *rec* | 31.78 ±0.98 | 25.38 ±1.27 | 36.5 |
| *ef1a* | 30.66 ±1.17 | 32.21 ±1.06 | 37.9 |

Figure S1. **Mitochondrial enrichment was confirmed for endogenous REC and XLF1 but not for other repair factors. (A)** Endogenous SPN-B, SPN-D, MRE11, and NBS tagged with Halo did not show mitochondrial enrichment in *Drosophila* egg chambers. The mitochondrial network and DNA were stained with anti-ATP5A antibody and DAPI, respectively. Oocyte nuclei are highlighted by yellow-dashed lines. Scale bar: 10 µm. **(B)** Representative images of endogenous HDM and XLF1 tagged with Halo in *Drosophila* egg chambers. XLF1 showed a weak mitochondrial signal, whereas HDM expression was too low to detect. The mitochondrial network and DNA were stained with anti-ATP5A antibody and DAPI, respectively. Oocyte nuclei are highlighted by yellow-dashed lines. Scale bar: 10 µm. **(C)** Endogenous REC tagged with Halo in egg chambers from the germarium up to stage 6 (top) or a stage 9 egg chamber (bottom), co-stained for the mitochondrial network (anti-ATP5A) and nuclei (DAPI). Extensive mtDNA replication occurs from region 2B of the germarium into later stages (Hill et al., 2014). REC in oocyte nuclei was not detected after stage 6 but remains mitochondrial throughout oogenesis. Fig. 1 D is a zoomed-in view of the stage 5/6 egg chamber shown in the top panel. **(D)** Sequence details of *rec* mutants generated by CRISPR/Cas9-based editing. Only one protein isoform of REC has been identified in *Drosophila*, and it was predicted by MitoProt to contain a putative mitochondrial targeting sequence at the N terminus (MTS, 2–44 amino acids), ZF: zinc finger, MCM: minichromosome maintenance domain. *rec*$^{\Delta MTS}$ contains a 132 bp deletion at the N-terminus, and *rec*$^{KO}$ contains a 130 bp deletion that introduces a stop codon at amino acid 90. **(E)** Immunoblot of mitochondrial fraction (Mito, anti-COX4) and whole cell lysate (WCL, cytoplasm: anti-tubulin, nucleus: anti-PCNA) of *REC-Halo*, *REC*$^{\Delta MTS}$-*Halo* and *REC*$^{KO}$-*Halo* ovaries. This confirmed that *REC*$^{\Delta MTS}$-*Halo* flies had a much-reduced mitochondrial REC and that REC expression was not detected in *REC*$^{KO}$-*Halo* flies. **(F)** Ct values from measuring *rec* and *ef1a* mRNA levels by RT-qPCR in Fig. 1 G (*n* = 5 biological replicates). Data represent mean ± SD. Source data are available for this figure: SourceData FS1.

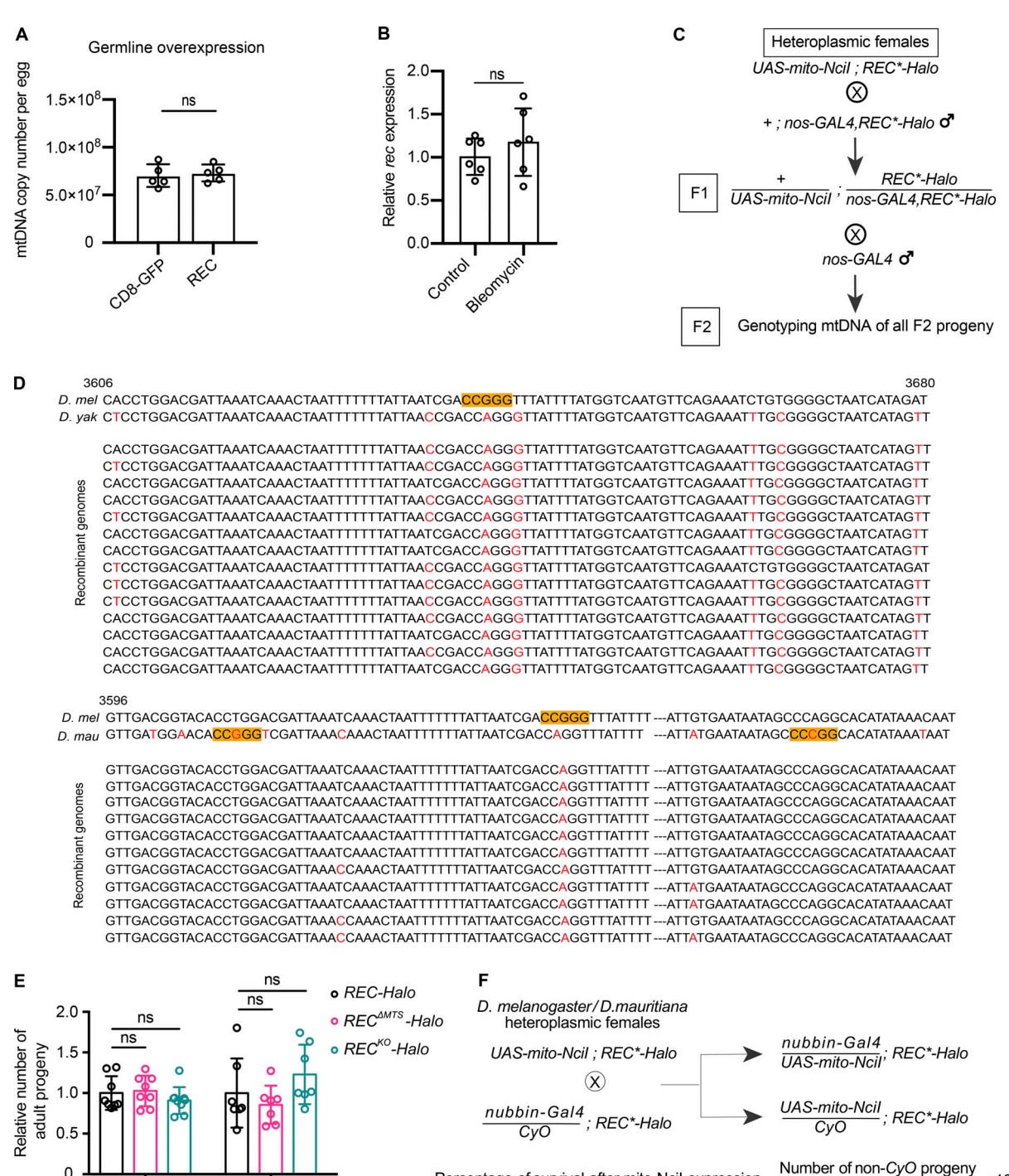

Figure S2. **REC is not essential for mtDNA replication but is required for recombination-based repair in mitochondria. (A)** mtDNA copy number per egg from females with germline overexpression of a control construct (CD8-GFP alone) or REC-GFP driven by germline *nos-GAL4* measured by qPCR (*n* = 5 biological replicates). Data represent mean ± SD, Student's one-sided *t* test, ns P > 0.05. **(B)** Relative *rec* mRNA levels in fly ovaries measured by RT-qPCR after treatment with bleomycin or PBS alone (control), normalized to *ef1a* (*n* = 6 technical replicates). Data represent mean ± SD, Student's one-sided *t* test, ns P > 0.05. **(C)** The cross scheme for the mtDNA recombination assay in the germline. *REC*-Halo* is used to represent that the crosses were performed in parallel with either *REC-Halo*, *REC^{ΔMTS}-Halo*, or *REC^{KO}-Halo* flies. **(D)** Sequences of recombinant mitochondrial genomes isolated from the two heteroplasmic fly lines. Highlighted base pairs indicate the NciI recognition sites in the parental genomes. **(E)** Relative numbers of adult progeny produced by *REC-Halo*, *REC^{ΔMTS}-Halo*, and *REC^{KO}-Halo* females that were either homoplasmic for *mt:NciI^{resistant}* or heteroplasmic for *D. melanogaster* and *D. mauritiana* mtDNA (*n* = 7–8 biological replicates). Females were crossed to *nos-GAL4* males and the number of adult progeny from eggs laid in the first 10 d was counted. Data represent mean ± SD, one-way ANOVA with Dunnett's multiple comparisons; ns, P > 0.05. **(F)** The cross scheme for the mtDNA recombination assay in the soma. The cross generates two types of sibling flies: one expressing mito-NciI under *nubbin-GAL4* and the other not expressing mito-NciI as *nubbin-GAL4* is replaced by the *CyO* balancer. The number of these two types of adult flies produced from each cross, which ranged between 37 and 162 flies was counted, and 4–5 crosses were quantified per line.

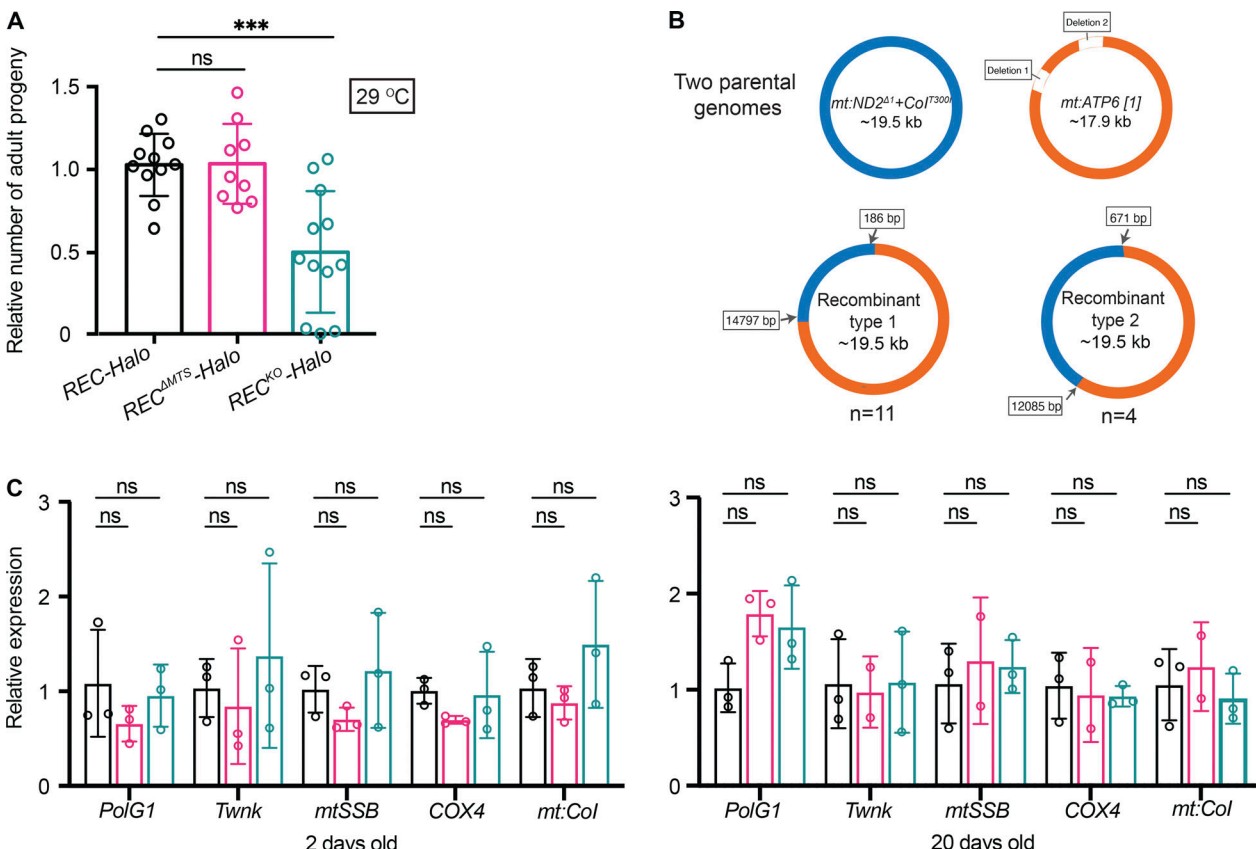

Figure S3. **REC mediates spontaneous recombination and has little impact on the expression of mitochondrial proteins during ageing. (A)** Relative numbers of adult progeny produced by *REC-Halo*, *REC[ΔMTS]-Halo* or *REC[KO]-Halo* females that were heteroplasmic for *mt:ATP6[1]* and *mt:ND2[Δ1]+CoI[T300I]* (n = 9–11 biological replicates) at 29°C. Females were crossed to *w1118* males and the number of adult progeny produced in the first 10 d was counted. The reduced female fertility of *REC[KO]-Halo* heteroplasmic flies could be caused by high levels of the temperature-sensitive mutant *mt:ND2[Δ1]+CoI[T300I]* (∼80%) and increased demand for REC as the meiotic recombination rate increases at higher temperatures (Grell, 1973; Altindag et al., 2020). Data represent mean ± SD, one-way ANOVA with Dunnett's multiple comparisons; ns, P > 0.05; ***, P < 0.005. **(B)** Maps of two types of recombinant mtDNA isolated from *REC-Halo* and *REC[ΔMTS]-Halo* flies heteroplasmic for *mt:ATP6[1]* and *mt:ND2[Δ1]+CoI[T300I]* (Fig. 4 A). The *mt:ATP6[1]* genome lacks several tandem repeats in the non-coding region, and thus is ∼1.6 kb shorter than the *mt:ND2[Δ1]+CoI[T300I]* genome (Ma and O'Farrell 2015). **(C)** Relative expression of nuclear and mtDNA genes encoding mitochondrial proteins in heads of adult males of 2 or 20–30 d old measured by RT-qPCR, normalized to *ef1a* (n = 3 biological replicates). Data represent mean ± SD, one-way ANOVA with Dunnett's multiple comparisons; ns, P > 0.05.

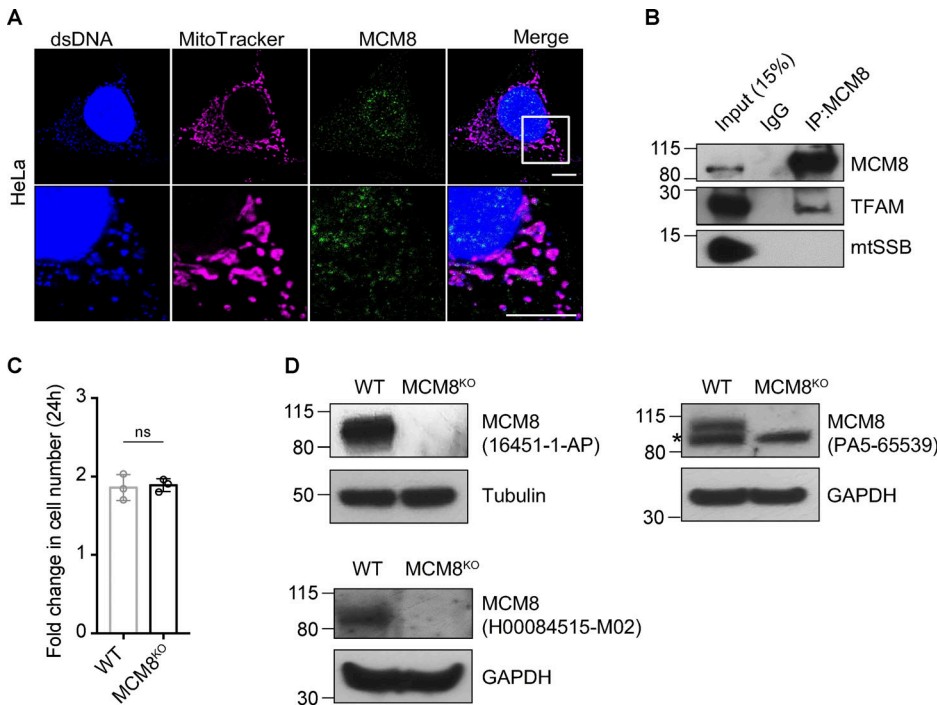

Figure S4. **MCM8 co-localizes with mitochondria and mtDNA in human cells. (A)** Representative images of HeLa cells showing co-localization of MCM8 with mitochondria and mtDNA using a different antibody to the one used in Fig. 5 A. Scale bars: 10 μm. **(B)** Co-immunoprecipitation assays using an anti-MCM8 antibody suggest that MCM8 interacts with TFAM, but not with mtSSB, in HEK293T cells. **(C)** The fold increase in cell numbers of wild-type and MCM8[KO] cells cultured in glucose media for 24 h ($n$ = 3 biological replicates). Data represent mean ± SD, Student's one-sided $t$ test; ns, P > 0.05. **(D)** Immunoblots of wild-type and MCM8[KO] cells confirm that the three MCM8 antibodies used in this study recognize the human MCM8 protein. Tubulin or GAPDH was blotted as the loading control. Source data are available for this figure: SourceData FS4.

**Provided online is Table S1. Table S1 a list of candidate genes examined for the mitochondrial enrichment of their protein products when overexpressed in *Drosophila* S2R+ cells (Fig. 1 A).**

