## [Peer Review File · The Journal of Cell Biology]

REC drives recombination to repair double-strand breaks in animal mtDNA

Anna Klucnika, Peiqiang Mu, Jan Jezek, Matthew McCormack, Ying Di, Charles Bradshaw, and Hansong Ma

Corresponding Author(s): Hansong Ma, Wellcome/Cancer Research UK Gurdon Institute

Review Timeline:

Submission Date:	2022-01-26
Editorial Decision:	2022-03-07
Revision Received:	2022-09-09
Editorial Decision:	2022-10-07
Revision Received:	2022-10-14

Monitoring Editor: Jodi Nunnari

Scientific Editor: Lucia Morgado-Palacin

Transaction Report:

DOI: <https://doi.org/10.1083/jcb.202201137>

March 7, 2022

Re: JCB manuscript #202201137

Dr. Hansong Ma
Wellcome/Cancer Research UK Gurdon Institute
Tennis Court Road
Cambridge CB2 1QN
United Kingdom

Dear Dr. Ma,

Thank you for submitting your manuscript entitled "The meiotic helicase REC drives recombination to repair double-strand breaks in animal mtDNA". The manuscript has been evaluated by expert reviewers, whose reports are appended below. Unfortunately, after an assessment of the reviewer feedback, our editorial decision is against publication in JCB.

As you will see, while the reviewers voice enthusiasm for the premise of your work, they also raise a number of major overlapping concerns. In particular, they feel that the main conclusions are not adequately supported by the data so request more evidence on the mitochondrial localization of REC and on its function in mtDNA recombination (see reviewers #1 and #3 specific comments on these points). In addition, there are several serious technical shortcomings that have been flagged by all the reviewers, and, specially, by the reviewer #2. The reviewer #3 also asks for the significance of MCM8 in mtDNA mutations and for the role of MCM9 in mtDNA repair in mammalian cells, particularly in the light of the minimal effects of single MCM8 KO. Given that the reviewers' concerns questions to some degree the core of the study and based on the extent of revisions that would be necessary to address their concerns, we cannot consider your manuscript for publication in JCB at this time. If you wish to expedite publication of the current data, it may be best to pursue publication at another journal - we would be happy to transfer your manuscript and reviews to the journal of your choice.

However, given interest in the topic, we would be open to an appeal of this decision and resubmission to JCB of a significantly revised and extended manuscript that completely addresses each of the reviewers' concerns in full. If you would like to resubmit this work to JCB, please contact the journal office to discuss an appeal of this decision or you may submit an appeal directly through our manuscript submission system. Discussion of a revision plan (possibly with reviewer input) can be beneficial to avoid time- and effort-consuming revisions that may not be sufficient for a successful resubmission to the journal. Please note that priority and novelty would be reassessed at submission and the paper would, of course, be subject to re-review by the same reviewers (if possible).

Regardless of how you choose to proceed, we hope that the comments below will prove constructive as your work progresses. We would be happy to discuss the reviewer comments further once you've had a chance to consider the points raised in this letter.

You can contact the journal office with any questions, cellbio@rockefeller.edu.

Thank you for thinking of JCB as an appropriate place to publish your work.

Sincerely,

Jodi Nunnari, Ph.D.
Editor-in-Chief
The Journal of Cell Biology

Lucia Morgado-Palacin, PhD
Scientific Editor
Journal of Cell Biology

Reviewer #1 (Comments to the Authors (Required)):

In this manuscript, Klucnika et al report unexpected roles of a meiotic helicase, REC in mitochondrial genome repair and maintenance. They show that, besides the nucleus, REC also localizes to mitochondria and interacts with mtDNA replication factors. REC is further enriched in mitochondria upon the induction of double-stranded DNA breaks. Both the REC knock-out and a deletion mutant, which is defective in mitochondrial localization, show age-associated decline of mobility and increase of

mtDNA variants. The frequency of mtDNA recombination induced by mito-REs is markedly reduced in these backgrounds.

It appears to be an interesting study on the first reading. However, after closely scrutinizing the data, I noticed that the evidence supporting several claims is not fully robust. Considering the importance of the question-whether there is homologous recombination in mitochondria, and its potential impact on the field, I feel that additional evidence is needed to support two key findings of this study.

1/ Does REC truly localize to the mitochondrial matrix? While REC appears to be associated with mitochondria in both the fluorescence imaging and the subcellular fraction experiments, there is no data showing that REC resides inside the mitochondrial matrix, which is a prerequisite for REC to be involved in mtDNA recombination. Images in Figure 1B & D are not clear, and many MCM8 puncta are outside of mitochondria in Figure 4A. The author could try the GFP complementation assay (Vacario et al, 2019 Cell Death and Disease; Ozawa et al, 2003 Nature Biotechnology) to confirm whether REC localizes inside the matrix.

2/ Does REC truly promote mtDNA homologous recombination? The author carried out a genetic test outlined in Figure S2C to indirectly assay the recombination frequency in germline. However, it is not stated in the legend or the Method section how the recombination frequency plotted in the Figure 3B was calculated. Does it indicate the percentage of F1 flies that produced viable adult F2 flies? Were there any F1 REC MTS-Halo, or F1 RECKo-Halo flies that laid eggs, but their eggs failed to hatch or develop to adult flies? It has been shown that mutations in several DNA repair/recombination factors such as DmRad51/SpnA and Blm cause maternal-effect embryonic lethality. If eggs of REC MTS-Halo or F1 RECKo-Halo flies had reduced viability, the actual recombination frequency in REC MTS-Halo or F1 RECKo-Halo background might be underestimated. The same concern also applies to the data in Figure 3C, and it should be addressed in both cases.

In Figure 2D, the authors show that REC MTS-Halo and F1 RECKo-Halo flies laid similar number of eggs as the control (REC-Halo). Did eggs of 3 different genotypes have comparable hatching rate? How about their development into adulthood? The same experiment should also be conducted in heteroplasmic flies of 3 different genotypes, with or without the expression of Mito-REs.

Other comments:

3/ Figure 2A. I find the result of co-immunoprecipitation not convincing, especially after looking into the uncropped blots. There are bands, though weaker in no-FLAG antibody control (SSB vs REC), no lysate control (TFAM vs REC) and everywhere (TAMAS vs REC). REC binds to ssDNA and is not required for mtDNA replication. Why would one expect that REC binds to mtDNA replication factors constitutively?

4/ Figure 3A. It is plausible that REC in the matrix might be recruited to DSBs on mtDNA. However, I am confused why and how Bleomycin could induce the import of REC into mitochondria. REC has a MTS, which often contains positively charged basic amino acid residues. Bleomycin can induce oxidative damages in mitochondria, and presumably would depolarize mitochondria and impair the import of MTS-bearing mitochondrial proteins.

In Figure 1E, REC localizes both in mitochondria and the nucleus, and appears equally in these two compartments. If there is 50% increase of REC in mitochondria after Bleomycin treatment as shown in Figure 3A, should we expect 50% decrease of REC in the nucleus? However, Bleomycin causes DSBs on both mtDNA and the nuclear DNA, and it is known that DSBs can induce the translocation of DNA repair proteins into nucleus. If REC protein level is increased in both mitochondria and the nucleus, where does these additional REC come from? Given there is no increase of REC transcript, are authors suggesting a translational boost of REC in response to DSBs?

5/ Figure 4D. What is the system error rate of mtDNA sequencing procedure used in Figure 4D and Figure 5E? Why the authors arbitrarily choose two different cut-off value in these two experiments? The higher number of variants in 20-30 days REC MTS-Halo or RECKo-Halo flies seems caused by one outlier in each dataset. Are the differences between REC and REC MTS or RECKo statistically significant? Are these seemingly minor differences responsible for the drastic decline of mobility observed in 20-day old REC MTS or RECKo flies?

I understand the purpose of these experiments, trying to explore REC function in mitochondria under normal, physiological condition (without the induction of DSBs by Mito-Res). However, the data in Figure 4D and 5E is not convincing. Perhaps, they could test whether REC is required for mtDNA recombination in a heteroplasmic line under metabolic stress showed in a previous study by the senior author (Figure 2, Ma and O'Farrell, 2016 eLife)?

Reviewer #2 (Comments to the Authors (Required)):

The mammalian MCM8 and MCM9 proteins form a complex with putative DNA helicase activity. The complex is implicated in homologous recombination, in both mitotic and meiotic cells. In the present report, the authors propose that MCM8 and MCM9

enter mitochondria and catalyzes homologous recombination, and repair of double strand DNA breaks in the organelle. The results are at odds with the majority previously published studies, which argue against homologous recombination and double-stranded break (DSB) repair in mitochondria. Previous reports have demonstrated that selective introduction DSBs into mutant mtDNA leads but to rapid degradation of linear mtDNA molecules. The absence of DSB repair explains why mutant mtDNA can be specifically eliminated by mitochondrially targeted zinc finger-nucleases in cell lines and mouse models.

The findings in the manuscript are therefore new and unexpected. Unfortunately, there are several experimental shortcomings, which causes me to question the validity of the presented data.

#1. The antibodies used in this study have not been properly validated. In fact, the experiments presented cause me to question if they actually recognize the MCM8 and MCM9 proteins. In figure 4B, there is not MCM8 observed in the nucleus. In addition, the levels of MCM8 and MCM9 does not appear to correlate. The signal for MCM9 is strong in the nucleus and weaker in mitochondria. In contrast, the MCM8 signal is relatively strong in the mitochondria, but absent in the nucleus. These two proteins are expected to form a complex, but seem to be present in very different concentrations in different cellular compartments. Do the antibodies recognize the correct proteins? The authors also need to verify that the antibodies used specifically recognize the expected proteins, MCM8 and MCM9. To this end, they should use RNAi and demonstrate the loss of immunoblot signals.

#2. Subcellular fractionation experiments should be presented in a better way. No size markers are available and the authors need to display larger section of the immunoblots. Are there many different bands present and is the interpretation of the immunoblot unambiguous?

#3. The immunostaining experiments are of poor quality. The authors need to improve image analysis and quantify data do demonstrate colocalization. Better resolution is required, and more details of imaging acquisition and processing should be provided. The authors should also include a positive control.

#4. MCM8 and MCM9 have not been identified in previous studies of the mitochondrial proteome (see mitocarta 3.0). The authors therefore need to clearly demonstrate import of the proteins into mammalian mitochondria, using labelled proteins and isolated mitochondria. The experiment should conclusively demonstrate membrane potential dependent import and cleavage of the N-terminal, mitochondrial targeting sequence.

#5. Immunoprecipitation experiments are not of sufficient quality to draw any conclusions about interactions between REC/MCM8 with the mitochondrial replication machinery.

In figure 2A, the authors only load 1% of the input as a control. The precipitated proteins, mtSSB, TFAM and POLG are overexpressed, which makes unspecific interactions very likely. Much less than 1% of REC is precipitated together with mtSSB, strongly questioning the idea of direct interactions. The authors need to use additional controls, including beads coated with a non-specific antibody. They need to quantify their data. They also need to demonstrate direct interactions between endogenous REC and mtSSB/TFAM (not overexpressed, tagged protein). In addition, the idea of direct interactions between REC/MCM8 and mitochondrial replication proteins should be demonstrated using at least one additional, biochemical assay. Interactions between purified proteins could for instance be analyzed by microscale thermophoresis.

#6. Protein expression in REC Δ MTS -Halo flies are much lower than in REC-Halo files (FigS1E). Apparently, loss of the predicted MTS-region destabilizes the protein, making direct comparisons difficult. In addition, a similar sized band is observed in both the mitochondrial and whole cell lysate, causing me to question of the "mitochondrial isoform" of REC is really lost. And if the REC-Halo really correspond to the mitochondrial isoform, why is this shorter, mitochondrial isoform of the protein not observed I the REC-Halo flies? The authors need to provide crisp data that clearly demonstrating the loss of the mitochondrial isoform upon deletion of the MTS.

Reviewer #3 (Comments to the Authors (Required)):

In this manuscript, Klucnika and co-workers characterized a previously unknown role for the meiotic helicase REC in mediating homologous recombination-based mitochondrial DNA repair in vivo in *Drosophila*, both in the germline and the soma. The topic of mitochondrial DNA repair has been debated for a long time, with experimental proof regarding active base excision repair machinery in the organellar matrix only. The existence of other sources of DNA repair, like HR and NHEJ mediated double-stranded break repair, is still debated. Several members of the nuclear HR and NHEJ machinery have been identified in mitochondria, but experimental evidence of their functionality is minimal.

The author's work fits very well in this ongoing debate. Characterizing the role of REC proteins in mtDNA repair provides experimental evidence to fill our knowledge gaps partially. The authors use directed genetics and a candidate screen in *drosophila* cells to identify previously unnoticed mitochondrial localization for the proteins REC and XLF. The authors analyze the protein REC and provide observational and functional validation of its role in mediating mtDNA repair after DSBs. Overall, I

found the manuscript presented technically and scientifically sound, with some exceptions described below. The topic is of great interest to the mitochondrial DNA research community and -- given the importance of mtDNA repair and mtDNA dysfunction in human diseases -- beyond. Therefore, I find it suitable for the readers of JCB.

I recommend considering the manuscript for publication in JCB as a report. Nevertheless, this is contingent on addressing a few major experimental points. Additionally, there are several minor concerns that I would like the authors to manage and or discuss, even for a report format.

My major concerns are:

Localization studies regarding REC. Although the imaging and fractionation approach shown in the text and supplementary are convincing, the authors denote a residual localization of REC in the mitochondria compartment, even after deleting the putative MTS. Given the partial phenotypes observed, it would be crucial to tackle this observation and discuss it further.

- Are the partial phenotypes due to residual localization or other repair players (like XLF1, which is rapidly abandoned?). If the phenotypes are due to the residual localization, I consider a key point to show which parts of REC sequences mediate the import. Is there a cryptic MTS? I would suggest the author design a set of partial REC constructs fused to Halo that can be used in cells to clarify the partial import.

Functional validation of REC in mtDNA HR in *Drosophila*. The experiments presented in Fig 3 by the authors are a solid experimental base showing a functional role for REC in mediating recombination mediated repair of mtDNA.

- In figure 3B, the effects of deleting REC MTS are partial. If (as above, and as stated by the authors) the partial localization is responsible for this, it becomes essential to address it. Irrespective, there are remaining recombination events in total REC KO, and the authors do not discuss this further. Are there additional players in mtDNA repair in *Drosophila*? Does XLF1 mediate the remaining recombination? Given that only two candidates were identified, I would expect the role of XLF to be further explored in the manuscript or at least discussed in the context of the results.

- Figure 3C provides additional experimental evidence for the role of REC in mediating the essential repair of mitochondrial DNA. The experimental approach used for this is not sufficiently explained and challenging to interpret, even with the schematic provided in the supplemental material. It is not clear the total number of flies analyzed per cross and how Ncil expression is determined (if it is). The formula takes into consideration only CyO to determine the survival rates. Are we sure that Ncil is still expressed in the mitochondria of those flies? Is there sequencing data showing the product of recombination for this particular experiment? In general, the data presented in 3C need would benefit from the additional description, with more inclusive captions.

- As for 3B, in 3C it is very puzzling that a deltaMTS-REC yields the same survival rate as a wild-type REC, reinforcing the idea of testing the role of XLF1.

- Would it be feasible to derive *drosophila* cell lines from the heteroplasmic flies (mel+yak or mel+mau) and perform a recombination assay under bleomycin treatment rather than exogenous restriction enzymes? In other words, can we hypothesize and support the idea that genotoxic stress (rather than the exogenous expression of RE) is inflicting DSB damage to the mtDNA that requires REC-mediated repair?

- Figure 3A: it is unclear what kind of quantification the authors are performing. Although there is mention of no increased transcription, the staining for REC in bleomycin conditions appears to be more intense. Is there a change in translation that could justify the increased localization? In general, this panel would benefit of a better explanation of the quantification strategies, additional representative images and a quantification of the marker fluctuations under different treatment, not only their relative level. For example, H₂O₂ treatment increases the mtSSB signal consistently: is this somehow masking an increased localization of REC?

Role of REC in mtDNA maintenance through repair and functional consequences in vivo. In figures 3D-E the authors provide direct evidence that deletion of REC increases the number of variants in mitochondrial DNA.

- While statistical analysis has been presented elsewhere, it is missing here. Is the data short of significance? This observation is very intriguing, especially when coupled with a decrease in climbing distance and ATP production.

- Which kind of variants were detected? Are these variants compatible with HR repair signatures?

- Do cells treated with bleomycin but deleted in REC show the same accumulation pattern?

Relevance of REC homolog MCM8 in human mitochondria. The authors conclude by analyzing the cellular localization and possible function of human REC homolog MCM8. This data is exciting and provides new functional roles for mitochondrial DNA repair proteins in the mammalian mitochondria. Nevertheless, I found it underdeveloped.

- Both MCM8 and MCM9 are analyzed, showing a robust mitochondrial localization. Even if *Drosophila* lacks an MCM9 ortholog, I find it quite important to investigate its role in the context of human analysis. The authors focused solely on MCM8 by performing Co-IP and targeted KO. I think the manuscript would enormously benefit by analyzing the effects of MCM9 KO in HeLa cells and possibly a double KO MCM8-MCM9. Although this might seem like a big undertaking, I think it would significantly increase our knowledge on mtDNA repair in mammalian mtDNA, especially in the light of the minimal effects of single MCM8 KO (as presented in S3B).

- I think the absence of a clear functional consequence of MCM8 deletions coupled with a detected increase in mtDNA mutations is counter-intuitive and needs to be discussed further. Are cells counter selecting the mutant variants when forced to grow on galactose? I would suggest looking at cellular viability and growth rate in the same conditions. MCM8 KO cells might indeed suffer in galactose due to the active counterselection in place, while a healthy membrane potential is maintained because

essential in those conditions.

- What is the mutation load of the cells cultured in S3B compared to Figure 4F?

While these were significant concerns, there are also several minor points that, if addressed, would significantly improve the manuscript:

1. In Figure 1B, HDM and XLF might be inverted as the images show a clear signal for HDM co-localizing with the mitochondria, while the text mention that HDM was too low to detect. Was the figure mislabeled?
2. The Co-IP analysis presented in Fig 2A and Fig 4C all focus on components of the mitochondrial replisome. Is it possible to show non-nucleoid proteins as well for control? In other words, do we have an unspecific binding of REC and MCM8 to all mitochondrial proteins?
3. In Figure 3A, there is no analysis of nuclear localization. Does REC increase its localization in the nuclear compartment, given the effects of bleomycin on both mt and nuclear genomes?
4. What is the reasoning for the statistical test choice in Suppl. S2? Two-way ANOVA would consider two variables, one being the KO in REC. What is the other variable? Did the authors explore further the changes in POLG1? Although not significant under this testing, it might be an interesting observation.

Detailed responses to reviewers' comments

Reviewer #1 (Comments to the Authors):

In this manuscript, Klucnika et al report unexpected roles of a meiotic helicase, REC in mitochondrial genome repair and maintenance. They show that, besides the nucleus, REC also localizes to mitochondria and interacts with mtDNA replication factors. REC is further enriched in mitochondria upon the induction of double-stranded DNA breaks. Both the REC knock-out and a deletion mutant, which is defective in mitochondrial localization, show age-associated decline of mobility and increase of mtDNA variants. The frequency of mtDNA recombination induced by mito-REs is markedly reduced in these backgrounds.

It appears to be an interesting study on the first reading. However, after closely scrutinizing the data, I noticed that the evidence supporting several claims is not fully robust. Considering the importance of the question-whether there is homologous recombination in mitochondria, and its potential impact on the field, I feel that additional evidence is needed to support two key findings of this study.

1/ Does REC truly localize to the mitochondrial matrix? While REC appears to be associated with mitochondria in both the fluorescence imaging and the subcellular fraction experiments, there is no data showing that REC resides inside the mitochondrial matrix, which is a prerequisite for REC to be involved in mtDNA recombination. Images in Figure 1B & D are not clear, and many MCM8 puncta are outside of mitochondria in Figure 4A. The author could try the GFP complementation assay (Vacario et al, 2019 Cell Death and Disease; Ozawa et al, 2003 Nature Biotechnology) to confirm whether REC localizes inside the matrix.

We would like to sincerely thank the reviewer for their interest in the topic and their insightful feedback. As they indicate, we have shown mitochondrial localisation of REC and MCM8 by both fluorescence imaging and the subcellular fraction experiments. The depletion of mitochondrial REC upon deletion of the N-terminal MTS is also strong evidence validating mitochondrial localisation. The reviewer highlights that REC and MCM8 are observed outside mitochondria in immunostainings. This is unsurprising because we also detect REC and MCM8 in Western blots of cytoplasmic fractions (Fig 1E and 5B). The reviewer rightly points out that we have not yet definitively shown that REC and MCM8 reside in the matrix. To address this, we performed proteinase K treatment of freshly isolated mitochondria with or without the swelling buffer to permeabilise the outer membrane and showed that REC and MCM8 are resistant to proteinase K digestion like other matrix proteins. We have added the new data in the revised version (Fig 1F and Fig 5C). We have also improved the image quality of Fig 1B and D by providing high-resolution pictures and zoomed-in views for key images.

2/ Does REC truly promote mtDNA homologues recombination? The author carried out a genetic test outlined in Figure S2C to indirectly assay the recombination frequency in germline. However, it is not stated in the legend or the Method section how the recombination frequency plotted in the Figure 3B was calculated. Does it indicate the percentage of F1 flies that produced viable adult F2 flies? Were there any F1 REC Δ MTS-Halo, or F1 RECko-Halo flies that laid eggs, but their eggs failed to hatch or develop to adult flies? It has been shown that mutations in several DNA repair/recombination factors such as DmRad51/SpnA and Blm cause maternal-effect embryonic lethality. If eggs of REC Δ MTS-Halo or F1 RECko-Halo flies had reduced viability, the actual recombination frequency in REC Δ MTS-Halo or F1 RECko-Halo background might be underestimated. The same concern also applies to the data in Figure 3C, and it should be addressed in both cases.

In Figure 2D, the authors show that REC Δ MTS-Halo and F1 RECko-Halo flies laid similar number of eggs as the control (REC-Halo). Did eggs of 3 different genotypes have comparable hatching rate? How about their development into adulthood? The same experiment should also be conducted in heteroplasmic flies of 3 different genotypes, with or without the expression of Mito-REs.

We can assure the reviewer that the hatching rate (egg-to-adult) for the three *rec* genotypes is not different. We apologize that we have not presented the data previously as part of Fig 2. We agree with the reviewer that it is an important piece of data and have added it to the revised manuscript (Fig 2C).

The germline mtDNA recombination frequency (Fig 3C and S2C) was calculated as the percentage of F1 mothers (i.e. heteroplasmic females expressing mito-Nci in their germline) that produced adult progeny with recombinant mtDNA. We genotyped any adult progeny by sequencing to confirm that their mtDNA is recombinant (Fig S2D). As the reviewer describes, for this quantification to be an accurate measure of mtDNA recombination frequency, the production of adult progeny must not be reduced in *rec* mutants independently of mtDNA recombination. Therefore, in the revised manuscript, we included two new sets of female fertility data to show that the reduced F2 progeny in REC^{KO} or REC ^{Δ MTS} flies is due to differences in mtDNA repair. First, we expressed mt-Ncil in flies that are homoplasmic for *mt:Ncil*^{resistant} and found that REC^{KO} or REC ^{Δ MTS} mutants produced a similar number of adult progeny as controls (Fig S2F). As mtDNA is not cut in these flies but mito-Ncil is still expressed, this controls for any fertility defects caused by REC^{KO} or REC ^{Δ MTS} from mito-Ncil expression that is not caused by mtDNA damages. Second, we measured the adult progeny number of flies that are heteroplasmic for mt:mel and mt:mau as suggested by the reviewer. We found that again *rec* mutants produced a

similar number of adult progeny as controls (Fig S2E), further demonstrating that the reduced adult progeny in our recombination assay is due to impaired mtDNA repair in *REC*^{KO} and *REC*^{ΔMTS} flies. In addition to the new fertility data and related statements, we added descriptions in Fig 3C and Fig S2C to explain how the recombination frequency in the germline was calculated in the revised manuscript.

The somatic mtDNA recombination frequency (Fig 3D and S2F) reflects the survival rate to adulthood of flies expressing mito-Ncil (i.e. straight winged) in their embryonic neuroblasts, ganglion mother cells and larval wing discs. The genetic cross (Fig S2F) generates two types of fly: one expressing mito-Ncil under nubbin-Gal4 and the other not expressing mito-Ncil under nubbin-Gal4. We then counted the number of these two types of adult flies that are produced from each cross. In Fig 3D, we have shown that adult survival is not reduced in *rec* mutant carrying *mt:Ncif*^{resistant} with mito-Nci expression, validating that the recombination assay in heteroplasmic lines only accounts for changes in survival due to differences in mtDNA repair by recombination. In the revised manuscript, we have added an extra description in Fig S2F legend.

Other comments:

3/ Figure 2A. I find the result of co-immunoprecipitation not convincing, especially after looking into the uncropped blots. There are bands, though weaker in no-FLAG antibody control (SSB vs REC), no lysate control (TFAM vs REC) and everywhere (TAMAS vs REC). REC binds to ssDNA and is not required for mtDNA replication. Why would one expect that REC binds to mtDNA replication factors constitutively?

mtDNA recombination factors may interact with mtDNA replication factors because replication is a necessary step of recombination. Stalled replication mtDNA forks are also a source of mtDNA DSBs. Furthermore, although there is no reduction in mtDNA copy number in REC mutants (Fig 2A), we cannot exclude that REC plays a non-essential role in mtDNA replication. We also reasoned that the co-IP experiment was interesting to pursue because any interaction between REC and mtDNA replication factors would further validate the mitochondrial matrix localisation of REC. We acknowledge the weak signal of our co-immunoprecipitation blots in Fig 2A of the original submission. REC is expressed at a very low level compared to many other mitochondrial proteins (Flybase <http://flybase.org/reports/FBgn0003227>, and modEncode transcriptome database <http://www.modencode.org/celniker/>). This has made the co-immunoprecipitation experiments using the endogenously tagged REC challenging and all antibodies suffered from non-specific binding. Given that these data are not relevant to our main conclusions, and we have demonstrated the matrix localisation of REC using the proteinase K protection assay (Fig 1F of the revised version), we have removed the data in original Fig 2A and related discussion from the revised manuscript.

4/ Figure 3A. It is plausible that REC in the matrix might be recruited to DSBs on mtDNA. However, I am confused why and how Bleomycin could induce the import of REC into mitochondria. REC has a MTS, which often contains positively charged basic amino acid residues. Bleomycin can induce oxidative damages in mitochondria, and presumably would depolarize mitochondria and impair the import of MTS-bearing mitochondrial proteins.

In Figure 1E, REC localizes both in mitochondria and the nucleus, and appears equally in these two compartments. If there is 50% increase of REC in mitochondria after Bleomycin treatment as shown in Figure 3A, should we expect 50% decrease of REC in the nucleus? However, Bleomycin causes DSBs on both mtDNA and the nuclear DNA, and it is known that DSBs can induce the translocation of DNA repair proteins into nucleus. If REC protein level is increased in both mitochondria and the nucleus, where does these additional REC come from? Given there is no increase of REC transcript, are authors suggesting a translational boost of REC in response to DSBs?

In the original manuscript, we showed that mitochondrial REC in ovaries increased after bleomycin treatment (Fig. 3A). Bleomycin has been previously shown to cause DSBs in both nuclear and mitochondrial DNA (Lim and Neims, 1987; Morel et al., 2008). Although we have not been able to find literature suggesting that bleomycin causes membrane depolarisation, bleomycin reduction could lead to ROS release (Mahmutoglu et al 1987), and oxidative damage may also be caused by dysfunctional mitochondrial function as a result of mtDNA DSBs induced by bleomycin. Unfortunately, we are unable to test if there is depolarisation of the mitochondrial membrane concurrent with REC enrichment to ovary mitochondria after bleomycin treatment because typical dyes used to do this (eg. JC-10 or TMRM) do not penetrate the tissue well.

We used imaging to measure the abundance of mitochondrial REC after treatment with DNA damaging agents because this is a direct and fast quantification. It is challenging to measure the nuclear abundance of REC with the same approach because this will be highly dependent on the plane of imaging. Moreover, we were most interested in the response to mtDNA damage, so did not further investigate the response of nuclear REC in the initial manuscript submission. However, we agree with the reviewer that it is pertinent to pursue and test whether there is a concurrent change in nuclear REC after bleomycin treatment. We therefore measured the change in REC levels by blotting subcellular fractions of ovarian tissues. This experiment showed that there is about 1.49 and 1.15 of fold increase of REC in mitochondrial and nuclear fractions, respectively, compared to the non-treated samples (Fig 3B). Our RT-qPCR suggested there was a similar level of increase in *rec* transcription after

the bleomycin treatment. It is not a significant increase due to large variations between different samples (Fig S2B), but might be sufficient to account for the increase in REC protein levels in the two compartments. The increased REC could also be caused by more translation of *rec* transcripts, or the additive effect of increased transcription and translation. It is hard to dissect this experimentally, so we have added a short statement in the revised manuscript to discuss these possibilities.

5/ Figure 4D. What is the system error rate of mtDNA sequencing procedure used in Figure 4D and Figure 5E? Why the authors arbitrarily choose two different cut-off value in these two experiments? The higher number of variants in 20-30 days REC Δ MTS-Halo or RECKo-Halo flies seems caused by one outlier in each dataset. Are the differences between REC and REC Δ MTS or RECKo statistically significant? Are these seemingly minor differences responsible for the drastic decline of mobility observed in 20-day old REC Δ MTS or RECKo flies?

I understand the purpose of these experiments, trying to explore REC function in mitochondria under normal, physiological condition (without the induction of DSBs by Mito-Res). However, the data in Figure 4D and 5E is not convincing. Perhaps, they could test whether REC is required for mtDNA recombination in a heteroplasmic line under metabolic stress showed in a previous study by the senior author (Figure 2, Ma and O'Farrell, 2016 eLife)?

We performed pair-end Illumina sequencing to detect rare mtDNA variants in flies and cells. Sequencing error rates and coverage variation can confound mtDNA sequencing data, making the detection of low-frequency mtDNA variants very challenging. 1% is often used by the field as the standard threshold to call a variant as heteroplasmy (e.g. Wei et al, Science 2019, Zhang et al, Science Advances, 2021), so we used a 1% cut-off for fly data. For the human data, we found many of our reads for variants below 3% show extreme/strong strand bias (reads being significantly overrepresented in one read direction, either forward or reverse, compared with the opposite read direction), indicating that they are likely to be false-positive SNPs (Guo et al, BMC Genomics 2012). Therefore, we used the 3% cut-off for human data. We have added our explanation for using a different threshold in the revised manuscript.

We observed an increased mtDNA mutation load in *rec* mutants concurrent with an increase in variation in mtDNA mutation load. Owing to the large variation, the mtDNA mutation load is not statistically significant between control and REC mutant flies. Nevertheless, the fact that the observed increase in mtDNA mutation load in aged REC mutants is not statistically significant does not mean it is not biologically significant. There is currently no way in which we could test whether these seemingly minor differences are responsible for the drastic decline in the healthspan of *rec* mutants. Indeed, it is a major scientific endeavour of the entire field of ageing to try and innovate technologies to directly show whether mtDNA mutations cause or are simply correlated with aging. This question is beyond what can be investigated in this study. However, the question is highly relevant, and to address the reviewer's concerns, we have re-write this part to state that increased mtDNA mutation level could contribute to reduced healthspan in aged *rec* mutants. As the reviewer mentioned outliers, we also replotted the figure by putting sequenced flies into categories to better reflect the distribution of populations with different mtDNA mutation levels for each age group (Fig 4B).

Sequencing could significantly underestimate the rate of mtDNA mutation *in vivo* because strong purifying selection will prevent the accumulation of newly generated detrimental mtDNA mutations from reaching a detectable level in organisms (Fan et al 2008, Stewart et al 2008, Freyer et al 2012, Ma et al 2014, Hill et al 2014). As suggested by the reviewer, we tested whether REC is required for spontaneous mtDNA recombination under metabolic stress using flies heteroplasmic for *mt:ATP6[1]* and *mt:ND2del1+ CoI/T300I*. To do this, we generated this heteroplasmic line and established 108 independent lineages for controls and *rec* mutants. As REC^{KO}-Halo flies showed reduced female fertility at 29°C (Fig S3A, this could be caused by high levels of the temperature sensitive mutant *mt:ND2^{ΔT}+CoI^{T300I}* (~80%) and increased demand for REC as the meiotic recombination rate increases at higher temperatures (Grell 1973, 1978; Ulku et al, 2020), we only compared the REC^{ΔMTS} flies with controls. We found that reducing mitochondrial REC is sufficient to diminish spontaneous recombination: Out of the 108 lineages we followed, 13 and 2 lineages with recombinant mtDNA were isolated from REC-Halo and REC^{ΔMTS}-Halo flies, respectively (Fig 4A). Therefore, we conclude that REC is also important for mtDNA recombination under metabolic stress. We believe this experiment further demonstrates that REC facilitates mtDNA recombination in both physiological conditions and upon the induction of DSBs (through mito-RE).

Reviewer #2 (Comments to the Authors):

The mammalian MCM8 and MCM9 proteins form a complex with putative DNA helicase activity. The complex is implicated in homologous recombination, in both mitotic and meiotic cells. In the present report, the authors propose that MCM8 and MCM9 enter mitochondria and catalyzes homologous recombination, and repair of double strand DNA breaks in the organelle. The results are at odds with the majority previously published studies, which argue against homologous recombination and double-stranded break (DSB) repair in mitochondria. Previous reports have demonstrated that selective introduction DSBs into mutant mtDNA leads but

to rapid degradation of linear mtDNA molecules. The absence of DSB repair explains why mutant mtDNA can be specifically eliminated by mitochondrially targeted zinc finger-nucleases in cell lines and mouse models.

The findings in the manuscript are therefore new and unexpected. Unfortunately, there are several experimental shortcomings, which causes me to question the validity of the presented data.

We would like to sincerely thank the reviewer for their assessment of our manuscript and useful, insightful feedback. As the reviewer rightly points out, mtDNA recombination is a contested subject because naturally occurring recombinant mitochondrial genomes could be very difficult to detect. The maternal inheritance and the clonality of mtDNA population make it hard to distinguish recombinant genomes from parental genomes. Moreover, individual organisms can easily carry >1 billion copies of mtDNA. Rare recombinants would be below the detection limit unless they have transmission advantages that allow them to reach a detectable level. As it is difficult to definitively demonstrate mtDNA recombination, the existence of mtDNA recombination has remained debatable. However, several examples of naturally occurring mtDNA recombinants have been reported in different animals, including salmon, flounders, carp, lizards, and mussels (Hoarau et al. 2002; X. Guo, Liu, and Liu 2006; Ujvari, Downton, and Madsen 2007; Ladoukakis and Zouros 2001; Strakova et al. 2016). Recombinant mtDNA has also been detected in patient tissues, mice and cell lines (Kraytsberg et al. 2004; Zsurka et al. 2005; Fan et al. 2012; D'Aurelio et al. 2004; Sato et al. 2005). We previously showed that mtDNA recombination occurs spontaneously in *D. melanogaster* at low frequencies and the induction of mtDNA DSBs by expressing mito-RE increases the frequency (Ma and O'Farrell 2015). We isolated various recombinant mitochondrial genomes and used them for mapping (Ma and O'Farrell 2016). Therefore, our fly system is unique and powerful setup as we can assay and quantify mtDNA recombination to interrogate the role of REC in mtDNA recombination.

#1. The antibodies used in this study have not been properly validated. In fact, the experiments presented cause me to question if they actually recognize the MCM8 and MCM9 proteins. In figure 4B, there is not MCM8 observed in the nucleus. In addition, the levels of MCM8 and MCM9 does not appear to correlate. The signal for MCM9 is strong in the nucleus and weaker in mitochondria. In contrast, the MCM8 signal is relatively strong in the mitochondria, but absent in the nucleus. These two proteins are expected to form a complex, but seem to be present in very different concentrations in different cellular compartments. Do the antibodies recognize the correct proteins? The authors also need to verify that the antibodies used specifically recognize the expected proteins, MCM8 and MCM9. To this end, they should use RNAi and demonstrate the loss of immunoblot signals.

Park et al (2013) showed that MCM8 and MCM9 co-immunoprecipitate and that MCM9 is more abundant in the nucleus than MCM8. Therefore, it has been previously established that the levels of MCM8 and MCM9 do not need to correlate for these proteins to form a functional complex. Antibodies previously used for MCM8 and MCM9 are hard to obtain as these were either raised for specific publications or discontinued (e.g. nb100-325). Therefore, in our original manuscript, we used two different polyclonal antibodies for each protein to validate the mitochondrial localisation of MCM8 and MCM9 by IF. Although slightly more nuclear MCM8 punctae were observed after immunostaining with MCM8 antibody PA5-65399 (Fig S3A) than the antibody used in (Fig 5A), the overall pattern is very similar: MCM8 is less abundant in the nucleus compared to MCM9. PA5-65399 was raised to a C-terminal region of MCM8 (undisclosed by manufacturer) while the other antibody was raised to the N-terminal region of MCM8, so there may be differential localisation of MCM8 isoforms. We can also not exclude the possibility that MCM8 and MCM9 may have uncharacterised functions either as a complex or independently.

The reviewer is completely right that it is important to show that antibodies used recognise MCM8 and MCM9. In the last few months, we immunoblotted wild-type and MCM8 KO cells with MCM8 antibodies used in this study and confirmed that they recognise the correct protein (Fig S4D and source data). In the meantime, an MCM8 antibody (16451-1-AP) previously used by three other publications (Liu et al, Cell Death Dis, 2020; Huang et al, Nau Commun, 2020; Zhang et al, Nat Commun 2020) became available during our revision time. We ordered it from Proteintech and verified it by immunoblotting the wild-type and MCM8 KO cells (Fig 5E, Fig S4D). As this antibody is more specific and gives a strong signal for WB, we re-performed the cell fraction assay using this antibody. This verified our previous observation using the PA5-65399 antibody that MCM8 was mainly found in the mitochondrial fraction, but also in the cytoplasm and nucleus (Fig 5B).

For MCM9, despite multiple attempts, we failed to generate an MCM9 mutant line that completely knocks out MCM9 protein. This is mainly because the HeLa Cas9 cells we used have multiple copies of MCM9-containing chromosome regions with heterogenous sequences, and only some copies were mutated at the gRNA-targeted sequences, creating "heterozygous/heterogenous" mutants. Given that we cannot verify the MCM9 antibody using a KO cell line, and *Drosophila* does not have the corresponding ortholog, we have decided to remove the MCM9 data from the manuscript.

#2. Subcellular fractionation experiments should be presented in a better way. No size markers are available and the authors need to display larger section of the immunoblots. Are there many different bands present and is the interpretation of the immunoblot unambiguous?

We have re-designed our figures to include large sections of all the immunoblots for the new Fig 1, Fig 5 and related supplementary Fig S4 in the revised version. We also uploaded all the source data to show whether other

bands are present. As the new MCM8 antibody (16451-1-AP) is very clean and sensitive, it provides strong and specific signals. Hence, all the new blots are much cleaner with fewer unspecific bands (see source data).

#3. The immunostaining experiments are of poor quality. The authors need to improve image analysis and quantify data to demonstrate colocalization. Better resolution is required, and more details of imaging acquisition and processing should be provided. The authors should also include a positive control.

To address the reviewer's concerns, we have provided images with higher resolutions. Given our cell fraction blotting data (Fig 5B, using two different MCM8 antibodies) strongly agree with our IF images regarding the MCM8 subcellular localisation (Fig. 5A), and we provide co-IP data demonstrating interactions between TFAM and MCM8, we do not think a positive control would add more to our conclusion.

#4. MCM8 and MCM9 have not been identified in previous studies of the mitochondrial proteome (see Mitocarta 3.0). The authors therefore need to clearly demonstrate import of the proteins into mammalian mitochondria, using labelled proteins and isolated mitochondria. The experiment should conclusively demonstrate membrane potential dependent import and cleavage of the N-terminal, mitochondrial targeting sequence.

MitoCarta, an inventory of mitochondrial proteins defined by an integration of different experimental approaches, only annotates 17 proteins as both mitochondrial and described by the GO term "DNA repair" in humans. It is now apparent that more mtDNA repair pathways are present, including recombination. As outlined above, we have shown mitochondrial localisation of REC, MCM8 and MCM9 by both fluorescence imaging and the subcellular fraction experiments. The depletion of mitochondrial REC upon deletion of the N-terminal MTS is also strong evidence validating mitochondrial localisation. What's more, MCM8 co-immunoprecipitated with TFAM in our reciprocal pull-down assays (Fig 5 and Fig S4), again demonstrating the mitochondrial localisation of MCM8. Many proteins listed in MitoCarta have been identified by these same experiments. However, we appreciate that it is very important to validate mitochondrial and matrix localisation of REC, MCM8 and MCM9. Therefore, in response to the reviewer's comments, we performed proteinase K treatment of isolated mitochondria and showed REC and MCM8 are resistant to proteinase K digestion after permeabilization of outer membrane, similar to other matrix proteins. Together, the imaging, co-immunoprecipitation, subcellular fractionation and proteinase K resistance data for REC and MCM8 conclusively show that these proteins localise to the mitochondrial matrix.

#5. Immunoprecipitation experiments are not of sufficient quality to draw any conclusions about interactions between REC/MCM8 with the mitochondrial replication machinery. In figure 2A, the authors only load 1% of the input as a control. The precipitated proteins, mtSSB, TFAM and POLG are overexpressed, which makes unspecific interactions very likely. Much less than 1% of REC is precipitated together with mtSSB, strongly questioning the idea of direct interactions. The authors need to use additional controls, including beads coated with a non-specific antibody. They need to quantify their data. They also need to demonstrate direct interactions between endogenous REC and mtSSB/TFAM (not overexpressed, tagged protein). In addition, the idea of direct interactions between REC/MCM8 and mitochondrial replication proteins should be demonstrated using at least one additional, biochemical assay. Interactions between purified proteins could for instance be analyzed by microscale thermophoresis.

We used co-immunoprecipitation experiments to investigate whether REC interacts with components of the mtDNA replisome. We always used endogenously tagged REC and PolG1 for our experiments. TFAM was tagged with GFP and inserted into second chromosome as a BAC construct such that its expression is controlled by the endogenous TFAM promoter. For mtSSB-GFP fly line, the mtSSB is under a ubiquitous driver, so it is an overexpression line. We acknowledge the weak signal of our co-immunoprecipitation blots in Fig 2A of the original manuscript. The endogenous REC is expressed at a very low level compared to many other mitochondrial proteins (Flybase <http://flybase.org/reports/FBgn0003227>, and modEncode transcriptome database <http://www.modencode.org/celniker/>). This has made the co-immunoprecipitation experiments using the endogenously tagged REC challenging and all antibodies suffered from non-specific binding. We thank the reviewer for their recommendation for additional controls and validation, but given that these data are not essential to our main conclusions, and we have demonstrated the matrix localisation of REC using the proteinase K protection assay (Fig 1F of the revised version), we have decided to remove the data in Fig 2A of the original submission and related discussion from the revised manuscript.

#6. Protein expression in REC Δ MTS -Halo flies are much lower than in REC-Halo flies (FigS1E). Apparently, loss of the predicted MTS-region destabilizes the protein, making direct comparisons difficult. In addition, a similar sized band is observed in both the mitochondrial and whole cell lysate, causing me to question of the "mitochondrial isoform" of REC is really lost. And if the REC-Halo really correspond to the mitochondrial isoform, why is this shorter, mitochondrial isoform of the protein not observed in the REC-Halo flies? The authors need to provide crisp data that clearly demonstrating the loss of the mitochondrial isoform upon deletion of the MTS.

From Western blots (Fig 1E) we observe that mitochondrial REC is depleted and there is no significant reduction in nuclear REC in REC Δ MTS-Halo flies (Fig 1E and S1E). However, we do not have any evidence to conclude that REC Δ MTS is destabilised. For Fig S1E, there was a reduced total loading of the whole sample for REC Δ MTS-Halo compared to REC-Halo in the Western blot of whole cell lysates of ovary samples REC-Halo and REC Δ MTS-Halo

flies (Fig S1E). This is visible from weaker bands from staining with Halo antibodies as well as control antibodies – most clearly PCNA and tubulin.

REC is a relatively large protein (it's predicted size is 97.6 kDa – which is increased by 33 kDa after Halo tagging). MitoProt software predicts that the first 44 amino acids of REC contain a putative MTS. It is possible that *in vivo* only the first few amino acids act as the MTS and are cleaved from the mitochondrial isoform. To maximise the reduction of mitochondrial REC, we deleted the whole 44 amino acids, which reduces the protein size by 4.7kDa. Resolving two large proteins with ≤ 4.7 kDa difference on gel is challenging. Given that we have verified the mitochondrial localisation of REC by imaging, subcellular fraction, and proteinase K protection assay, we believe there is already enough evidence to conclude that REC is imported into the mitochondrial matrix.

Reviewer #3 (Comments to the Authors):

In this manuscript, Klucnika and co-workers characterized a previously unknown role for the meiotic helicase REC in mediating homologous recombination-based mitochondrial DNA repair in vivo in *Drosophila*, both in the germline and the soma. The topic of mitochondrial DNA repair has been debated for a long time, with experimental proof regarding active base excision repair machinery in the organellar matrix only. The existence of other sources of DNA repair, like HR and NHEJ mediated double-stranded break repair, is still debated. Several members of the nuclear HR and NHEJ machinery have been identified in mitochondria, but experimental evidence of their functionality is minimal.

The author's work fits very well in this ongoing debate. Characterizing the role of REC proteins in mtDNA repair provides experimental evidence to fill our knowledge gaps partially. The authors use directed genetics and a candidate screen in *Drosophila* cells to identify previously unnoticed mitochondrial localization for the proteins REC and XLF. The authors analyze the protein REC and provide observational and functional validation of its role in mediating mtDNA repair after DSBs. Overall, I found the manuscript presented technically and scientifically sound, with some exceptions described below. The topic is of great interest to the mitochondrial DNA research community and -- given the importance of mtDNA repair and mtDNA dysfunction in human diseases -- beyond. Therefore, I find it suitable for the readers of JCB.

I recommend considering the manuscript for publication in JCB as a report. Nevertheless, this is contingent on addressing a few major experimental points. Additionally, there are several minor concerns that I would like the authors to manage and or discuss, even for a report format.

My major concerns are:

Localization studies regarding REC. Although the imaging and fractionation approach shown in the text and supplementary are convincing, the authors denote a residual localization of REC in the mitochondria compartment, even after deleting the putative MTS. Given the partial phenotypes observed, it would be crucial to tackle this observation and discuss it further.

- Are the partial phenotypes due to residual localization or other repair players (like XLF1, which is rapidly abandoned?). If the phenotypes are due to the residual localization, I consider a key point to show which parts of REC sequences mediate the import. Is there a cryptic MTS? I would suggest the author design a set of partial REC constructs fused to Halo that can be used in cells to clarify the partial import.

We would like to sincerely thank the reviewer for their evaluation of our manuscript and considered feedback. As the reviewer highlights, we used imaging and western blots of subcellular fractions (Fig 1D&E) to show that mitochondrial REC is depleted - but not abolished - in the *REC^{ΔMTS}-Halo* line. We also validated the importance of the N-terminus sequence of REC in mediating its mitochondrial import in S2 cells by showing that mitochondrial enrichment of REC-GFP was no longer observed after the deletion of the N-terminus. It may be that there are additional amino acids either at the termini or internally within the protein that may contribute to mitochondrial localisation. However, our REC-GFP cell system based on imaging is not sensitive enough to detect or dissect which other regions are responsible for the low residual level of mitochondrial REC. We did not pursue longer truncations in the fly because this will likely affect the nuclear function of REC and so diminish the value of the *REC^{ΔMTS}-Halo* line. It is also clear that REC is not the only driver of mitochondrial recombination because recombinant mtDNA was still formed in *REC^{KO}-Halo* flies (Fig 3C, D). It may be that other mitochondrial helicases also function to drive recombination, or REC-independent gene conversion is responsible for a proportion of DBS repair in mitochondria. It is unlikely that XLF1 is responsible for the remaining recombinants in *REC^{KO}* flies because XLF1 mediates NHEJ which is an error-prone repair mechanism that leads to indels at breakpoints, whereas mtDNA sequencing confirmed that accurate crossovers were obtained from all lines and so were therefore generated by homology-dependent recombination. We hope that our data acts as a stimulus and guide for future studies identify other mediators of mitochondrial recombination. To address the reviewer's concerns, we include a discussion of the possible redundant mechanisms of recombination in the revised manuscript. We also pointed out that other signals may be directing partial mitochondrial localisation of *REC^{ΔMTS}* in the revised manuscript.

Functional validation of REC in mtDNA HR in *Drosophila*. The experiments presented in Fig 3 by the authors are a solid experimental base showing a functional role for REC in mediating recombination mediated repair of mtDNA.

We would like to express our appreciation for the reviewer's positive feedback on the use of the genetic system to study mtDNA recombination in *Drosophila*.

- In figure 3B, the effects of deleting REC MTS are partial. If (as above, and as stated by the authors) the partial localization is responsible for this, it becomes essential to address it. Irrespective, there are remaining recombination events in total REC KO, and the authors do not discuss this further. Are there additional players in mtDNA repair in *Drosophila*? Does XLF1 mediate the remaining recombination? Given that only two candidates

were identified, I would expect the role of XLF to be further explored in the manuscript or at least discussed in the context of the results.

We used three lines to study the function of REC: *REC*, *REC^{ΔMTS}* and *REC^{KO}*. The *REC^{ΔMTS}* line often showed an intermediary phenotype, and we hypothesise that is likely because there is some residual mitochondrial REC activity as we observe some mitochondrial REC in this line by Western blotting of subcellular fractionations. It is also plausible that the loss of the nuclear functions contributes to the more severe phenotypes observed *REC^{KO}* flies. In addition, REC is unlikely the only driver of mitochondrial recombination because recombinant mtDNA was still formed in *REC^{KO}-Halo* flies. It may be that other mitochondrial helicases also function to drive recombination. As mentioned above, it is unlikely that XLF1 is responsible for the remaining recombinants in *REC^{KO}* flies because XLF1 mediates NHEJ which is an error-prone repair mechanism that leads to indels at breakpoints, whereas mtDNA sequencing confirmed that accurate crossovers were obtained from all lines and so were therefore generated by homologous recombination. To address the reviewer's concerns, we have included a discussion of the potential cause of these different phenotypes and other possible mechanisms of recombination in the revised manuscript.

- Figure 3C provides additional experimental evidence for the role of REC in mediating the essential repair of mitochondrial DNA. The experimental approach used for this is not sufficiently explained and challenging to interpret, even with the schematic provided in the supplemental material. It is not clear the total number of flies analyzed per cross and how Ncil expression is determined (if it is). The formula takes into consideration only CyO to determine the survival rates. Are we sure that Ncil is still expressed in the mitochondria of those flies? Is there sequencing data showing the product of recombination for this particular experiment? In general, the data presented in 3C need would benefit from the additional description, with more inclusive captions.

The somatic mtDNA recombination frequency (Fig 3D and S2F) reflects the survival rate to adulthood of flies expressing mito-Ncil (i.e. straight winged) in their embryonic neuroblasts, ganglion mother cells and larval wing discs. The genetic cross (Fig S2F) generates two types of fly: one expressing mito-Ncil under nubbin-Gal4 and the other not expressing mito-Ncil under nubbin-Gal4. We then counted the number of these two types of adult flies that are produced from each cross which ranged between 37 to 162 flies and 4-5 crosses were quantified per line. Unlike for the germline recombination, most tissues of the flies remain heteroplasmic for parental mtDNA because mt-Ncil is only expressed in a small number of tissues by the nubbin-Gal4 driver (embryonic neuroblasts, ganglion mother cells and larval wing discs). Therefore, recombinant mtDNA will only make up a small proportion of total mtDNA and so we are unable to directly detect mtDNA recombination in somatic tissues by sequencing.

To show that mito-Ncil is expressed in the somatic tissues in this genetic cross and that this leads to lethality, we also performed the same quantification for flies carrying wild-type mtDNA containing Ncil recognition sites (*mt:mel*, *mt:yak* or *mt:mau*) and expressing mito-Ncil by the nubbin-Gal4 driver, and this resulted in just 0-3% adult survival in all lines. We also performed the same quantification for flies carrying *mt:Ncil^{resistant}* mtDNA and expressing mito-Ncil by the nubbin-Gal4 driver, which resulted in ~100% adult survival in all lines (shown in Fig 3C). This validates that mito-Ncil is expressed and that this results in lethality when mtDNA are cut but not when mtDNA cannot be cut (or are repaired by recombination).

It is apparent from the reviewer's comments that we did not describe the methodologies used in sufficient clarity and detail. We would like to apologise for the confusion. We have extended the description of the mtDNA recombination assays in figure legends and other relevant places to improve data presentation in the revised manuscripts.

- As for 3B, in 3C it is very puzzling that a deltaMTS-REC yields the same survival rate as a wild-type REC, reinforcing the idea of testing the role of XLF1.

We agree with the reviewer that it is intriguing that *REC^{ΔMTS}* does not result in a severe recombination reduction in somatic tissues, unlike the reduction observed in the germline. As discussed above, this could be due to the residual mitochondrial REC activity, or REC-independent repair pathways inside mitochondria. In the somatic recombination assay we cannot rule out that XLF1-mediated NHEJ is responsible for mtDNA repair because, unlike for the germline recombination assay, we are unable to sequence the repaired mtDNA. We therefore are yet unable to dissect the role of XLF1 in mitochondria. We hope that our data acts as a stimulus and guide for future studies identify other mediators of mitochondrial recombination and repair, including the role of mitochondrial XLF1. We have included a paragraph in the revised manuscript to discuss possible redundant mechanisms of mtDNA recombination.

- Would it be feasible to derive drosophila cell lines from the heteroplasmic flies (*mel+yak* or *mel+mau*) and perform a recombination assay under bleomycin treatment rather than exogenous restriction enzymes? In other words, can we hypothesize and support the idea that genotoxic stress (rather than the exogenous expression of RE) is inflicting DSB damage to the mtDNA that requires REC-mediated repair?

We performed Illumina sequencing to detect rare mtDNA variants to test the role of REC in mitochondrial repair and safeguarding under normal physiological conditions. However, as the reviewer points out, we could also test this using a system to induce and select recombinant mtDNA without exogenous expression of a mito-RE. We agree that this is important to investigate. Unfortunately, this is challenging to establish a system to assay mtDNA recombination in culture heteroplasmic cells. However, for the revised manuscript, we tested the role of REC in mtDNA recombination in a heteroplasmic fly line that allows the detection of mtDNA recombination without the need for exogenous mito-RE expression (i.e. spontaneous recombination) (Ma & O'Farrell 2016). This system uses flies heteroplasmic for *mt:ATP6[1]* – a functional mtDNA genome, and *mt:ND2del1+ CoIT300I* – a temperature-sensitive lethal mutant with a non-coding region that confers a selfish transmission advantage. In most cases, *mt:ND2del1+ CoIT300I* outcompetes *mt:ATP6[1]*, and the entire lineage dies at restriction temperature (29°C). However, in some lineages, spontaneous recombination generates mtDNA that carries the functional *mt:Col* from *mt:ATP6[1]* and the non-coding region from *mt:ND2del1+ CoIT300I*. These recombinants have a selective advantage over the two parental genomes as they combine the functional and transmission advantages, and thus take over. Furthermore, their appearance allows the lineages bearing them to survive at 29°C. Using this system, we found that reducing mitochondrial REC vastly diminished the frequency of isolating spontaneous recombinants. This experiment demonstrates that REC is also required for spontaneous mtDNA recombination under more physiological conditions.

- Figure 3A: it is unclear what kind of quantification the authors are performing. Although there is mention of no increased transcription, the staining for REC in bleomycin conditions appears to be more intense. Is there a change in translation that could justify the increased localization? In general, this panel would benefit of a better explanation of the quantification strategies, additional representative images and a quantification of the marker fluctuations under different treatment, not only their relative level. For example, H₂O₂ treatment increases the mtSSB signal consistently: is this somehow masking an increased localization of REC?

We used imaging to measure the abundance of mitochondrial REC after treatment with DNA damaging agents because this is a direct and fast quantification. To do this, we used mtSSB-GFP to define the region of interest (i.e. mitochondria) and then measured the average fluorescence intensity of Halo staining (i.e. mitochondrial REC-Halo). Image acquisition settings were kept consistent throughout. Therefore, the intensity of mtSSB-GFP fluorescence (above that required to threshold the region of interest) does not affect the Halo fluorescence measurements. To address the reviewer's concerns, we added extra descriptions in the figure legend and Materials and Methods section to clarify the descriptions.

Moreover, to confirm our quantification based on imaging, we also measured the change in REC amount by blotting subcellular fractions of ovarian tissues. This experiment showed that the bleomycin treatment increases the amount of REC by ~1.49x and 1.15x in mitochondrial and nuclear fractions, respectively (Fig 3B). Our RT-qPCR suggested a small increase in *rec* transcription after the bleomycin treatment. It is not a significant increase due to large variations between different samples (Fig S2B), but might be sufficient to account for the increase in REC protein levels in the two compartments. The increased REC could also be caused by more translation of *rec* transcripts, or the additive effect of increased transcription and translation. It is hard to dissect this experimentally, so we have added a statement in the revised manuscript to discuss these possibilities.

Role of REC in mtDNA maintenance through repair and functional consequences in vivo. In figures 3D-E the authors provide direct evidence that deletion of REC increases the number of variants in mitochondrial DNA.

- While statistical analysis has been presented elsewhere, it is missing here. Is the data short of significance? This observation is very intriguing, especially when coupled with a decrease in climbing distance and ATP production.

Owing to the large variation, the mtDNA mutation load is not statistically significant between control and REC mutant flies. However, the fact that the observed increase in mtDNA mutation load in aged REC mutants is not statistically significant does not mean it is also biologically insignificant. There is currently no way in which we could test whether these seemingly minor differences are responsible for the drastic decline in healthspan of REC mutants. However, the question is highly relevant, and we acknowledge that we did not sufficiently discuss it. To address the reviewer's comment, we have re-write this part to state that increased mtDNA mutation level could contribute to reduced healthspan in aged *rec* mutants. We also replotted the figure by putting sequenced flies into categories to better reflect the distribution of populations with different mtDNA mutation levels for each age group (Fig 4B).

- Which kind of variants were detected? Are these variants compatible with HR repair signatures?

Recombination is an accurate repair mechanism, so we expect to observe increased mtDNA mutations that are signatures of other mtDNA repair pathways in flies with compromised mtDNA recombination (i.e. *rec* mutants). Indeed, we observe an increase in the number of point mutations in *rec* mutants but did not observe insertions or deletions.

- Do cells treated with bleomycin but deleted in REC show the same accumulation pattern?

The reviewer suggests that we perform deep sequencing of mtDNA of *Drosophila* cells lacking REC to test whether these also show increased mtDNA mutation accumulation. It may be expected that more mtDNA mutations may be tolerated *in vitro* due to reduced reliance on oxidative metabolism. The mtDNA mutation load would be expected to also increase after treatment with a DNA damaging agent like bleomycin. However, bleomycin also causes nuclear damage which can itself lead to increased mtDNA mutation accumulation. In this case, it would be difficult to distinguish whether any observed effects are due to compromised mtDNA or nuclear repair.

Relevance of REC homolog MCM8 in human mitochondria. The authors conclude by analyzing the cellular localization and possible function of human REC homolog MCM8. This data is exciting and provides new functional roles for mitochondrial DNA repair proteins in the mammalian mitochondria. Nevertheless, I found it underdeveloped.

- Both MCM8 and MCM9 are analyzed, showing a robust mitochondrial localization. Even if *Drosophila* lacks an MCM9 ortholog, I find it quite important to investigate its role in the context of human analysis. The authors focused solely on MCM8 by performing Co-IP and targeted KO. I think the manuscript would enormously benefit by analyzing the effects of MCM9 KO in HeLa cells and possibly a double KO MCM8-MCM9. Although this might seem like a big undertaking, I think it would significantly increase our knowledge on mtDNA repair in mammalian mtDNA, especially in the light of the minimal effects of single MCM8 KO (as presented in S3B).

Despite multiple attempts, we failed to generate an MCM9 mutant line that completely knocked out MCM9 protein. This is mainly because the HeLa Cas9 cells we used have multiple copies of MCM9-containing chromosome regions with heterogenous sequences, and only some copies were mutated at the gRNA-targeted sequences, creating "heterozygous/heterogenous" mutants. Therefore, we only managed to obtain three hypomorphic alleles, which carry small indel or frameshift that resulted in premature stop codons in some of the MCM9 copies. As heterozygous mutations, they are not ideal to examine the effect of MCM9 on mtDNA maintenance. Without a KO, we also cannot verify the MCM9 antibody used for IF and WB in Fig 4 of the original submission. Given that we cannot verify the antibody and *Drosophila* does not have the corresponding ortholog, we have decided to remove the MCM9 data from the manuscript.

- I think the absence of a clear functional consequence of MCM8 deletions coupled with a detected increase in mtDNA mutations is counter-intuitive and needs to be discussed further. Are cells counter selecting the mutant variants when forced to grow on galactose? I would suggest looking at cellular viability and growth rate in the same conditions. MCM8 KO cells might indeed suffer in galactose due to the active counterselection in place, while a healthy membrane potential is maintained because essential in those conditions.

We think it is a good idea to investigate this more fully. Hence, we measured the cell growth rate, ATP levels, mtDNA copy number and mitochondrial membrane potential of MCM8 KO cells in both glucose and galactose at multiple timepoints. We found that MCM8^{KO} cells have similar growth rates, mtDNA copy numbers, and ATP levels as controls when cultured in standard media with glucose (Fig 5F, Fig S4C). When cultured in galactose media, MCM8^{KO} cells indeed show a much-reduced ATP production, whereas the total mtDNA copy number remains unchanged (Fig 5F). Their mitochondria also have a lower membrane potential after a longer incubation in galactose media (Fig 5F). These data, together with the increased mtDNA mutation levels observed with MCM8 KO cells, support our conclusion that MCM8 safeguards mtDNA to maintain mtDNA integrity and mitochondrial function.

- What is the mutation load of the cells cultured in S3B compared to Figure 4F?

The cells were sequenced after 22 generations of passage, whilst the JC-10 quantifications were performed on the same lines at passages 19 and 20. We will ensure that for future experiments the cells are from equivalent batches so that sequenced mtDNA mutation load is comparable throughout.

While these were significant concerns, there are also several minor points that, if addressed, would significantly improve the manuscript:

1. In Figure 1B, HDM and XLF might be inverted as the images show a clear signal for HDM co-localizing with the mitochondria, while the text mention that HDM was too low to detect. Was the figure mislabeled?

We apologise for the confusion caused by our unclear description. In the text, we wrote that "Endogenous HDM levels were too low to detect, and overexpressed HDM was not enriched in mitochondria (Fig. S1B)". Here we were referring to that endogenously tagged HDM-Halo could not be detected by imaging (Fig. S1B top image of the original submission). To overcome the low expression of endogenous Halo, we also generated a fly line overexpressing HDM-GFP. In this case, HDM-GFP was not found to be enriched in ovary mitochondria (Fig. S1B bottom image). This is in contrast to the extensive mitochondrial localisation observed when HDM-GFP was

overexpressed in *Drosophila* S2R+ cells in vitro (Fig 1B). We have removed the HDM-GFP overexpression data in the revised manuscript to avoid further confusion.

2. The Co-IP analysis presented in Fig 2A and Fig 4C all focus on components of the mitochondrial replisome. Is it possible to show non-nucleoid proteins as well for control? In other words, do we have an unspecific binding of REC and MCM8 to all mitochondrial proteins?

REC is expressed at very low levels, making co-immunoprecipitation experiments challenging as we used endogenously tagged REC. Given that these data are not essential to our main conclusions, we have removed the co-immunoprecipitation results shown in Fig 2A and related discussions from the revised manuscript.

Regarding Fig 4C of the original submission, we previously performed reciprocal pull-downs using both TFAM and MCM8 antibodies. We only presented the pull-down data using the TFAM antibody (Fig 4C in our original submission). In the revised manuscripts, we added the pull-down data using the MCM8 antibody (Fig S4B). In addition, we also probed mtSSB for both pulldowns previously, and we have provided this new piece of data in the current version too. The pull-down assay using the TFAM antibody suggests that TFAM interacts with both MCM8 and mtSSB (Fig 5D). The Co-IP assay using MCM8 antibody only pulled down TFAM. This suggests that MCM8 only interacts with TFAM but not mtSSB in physiological conditions. However, because the MCM8 antibody has not been tested for co-IP before, it may not work very efficiently for such an application to detect protein interactions, especially when the interaction is weak. Therefore, we cannot rule out the possibility that MCM8 could interact with mtSSB. Nevertheless, the interaction between TFAM and MCM8 is convincing given that it was revealed by both pull-downs (Fig 5D & Fig S4B).

3. In Figure 3A, there is no analysis of nuclear localization. Does REC increase its localization in the nuclear compartment, given the effects of bleomycin on both mt and nuclear genomes?

We used imaging to measure the abundance of mitochondrial REC after treatment with DNA damaging agents because this is a direct and fast quantification. As the reviewer points out, bleomycin has been previously shown to cause DSBs in both nuclear and mitochondrial DNA (Lim and Neims, 1987; Morel et al., 2008). It is challenging to measure the nuclear abundance of REC because this will be highly dependent on the plane of imaging. Hence, we performed subcellular fractionation of ovaries and measured the amount of REC in both mito and nuclear fractions using Western blot. This experiment showed that REC level increases by 1.49x in the mitochondrial fraction and 1.15x in the nuclear fraction after 2h of bleomycin treatment (Fig 3B). Therefore, bleomycin treatment increases the total amount of REC in ovaries, potentially via additional translation or transcription, or both. We have added new data and related discussion in the relevant section.

4. What is the reasoning for the statistical test choice in Suppl. S2? Two-way ANOVA would consider two variables, one being the KO in REC. What is the other variable? Did the authors explore further the changes in POLG1? Although not significant under this testing, it might be an interesting observation.

We used two-way ANOVA to consider the effect of mutations in REC and the gene of interest. We amended the statistical analysis in the revised manuscript to perform a one-way ANOVA to just consider the effect of mutations in REC. We also found the increased expression of PolG1 in REC mutants very interesting. PolG1 is a crucial component of the mtDNA DNA polymerase, but also contains an exonuclease domain that can mediate mtDNA degradation (Yu et al 2018, Nissanka et al 2018, Peeva et al 2018). Given that no changes in mtDNA copy number in *rec* mutants were observed, and no other replication components we tested showed a similar pattern as PolG1, we did not explore the observation for this manuscript. However, we agree with the reviewer that this observation is interesting and worth further validation and investigation.

October 7, 2022

RE: JCB Manuscript #202201137R-A

Dr. Hansong Ma
Wellcome/Cancer Research UK Gurdon Institute
Tennis Court Road
Cambridge CB2 1QN
United Kingdom

Dear Dr. Ma:

Thank you for submitting your revised manuscript entitled "REC drives recombination to repair double-strand breaks in animal mtDNA". Two of the original reviewers have now assessed your revised manuscript and, as you can see, they are satisfied with revisions. Thus, we would be happy to publish your paper in JCB pending final revisions necessary to meet our formatting guidelines (see details below).

A. MANUSCRIPT ORGANIZATION AND FORMATTING:

1) Text limits: Character count for Reports is < 20,000, not including spaces. Count includes title page, abstract, introduction, results, discussion, and acknowledgments. Count does not include materials and methods, figure legends, references, tables, or supplemental legends.

2) Figures limits: Reports may have up to 5 main text figures.

***** Please note that main text figures should be provided as individual, editable files.**

3) Figure formatting:

***** Molecular weight or nucleic acid size markers must be included on all gel electrophoresis. Please, include MWs in main Figs. 1E-F, 3B, 5B-E, and supplemental Figs. 4B, 4D.**

***** Scale bars must be present on all microscopy images, including inset magnifications. Please include scale bars in main Figs. 5A (inset magnifications) and supplemental Figs. 4A.**

Also, please avoid pairing red and green for images and graphs to ensure legibility for color-blind readers. If red and green are paired for images, please ensure that the particular red and green hues used in micrographs are distinctive with any of the colorblind types. If not, please modify colors accordingly or provide separate images of the individual channels.

4) Statistical analysis:

Error bars on graphic representations of numerical data must be clearly described in the figure legend.

***** The number of independent data points (n) represented in a graph must be indicated in the legend. Please, indicate 'n' for main Figs. 2A, 2B, 3C, and supplemental Fig. 4C.**

***** Please, also indicate in the legend whether 'n' refers to technical or biological replicates (i.e. number of analyzed cells, samples or animals, number of independent experiments).**

If independent experiments with multiple biological replicates have been performed, we recommend using distribution-reproducibility SuperPlots (please, see Lord et al., JCB 2020) to better display the distribution of the entire dataset, and report statistics (such as means, error bars, and P values) that address the reproducibility of the findings.

Statistical methods should be explained in full in the materials and methods in a separate section.

For figures presenting pooled data the statistical measure should be defined in the figure legends.

Please also be sure to indicate the statistical tests used in each of your experiments (both in the figure legend itself and in a separate methods section) as well as the parameters of the test (for example, if you ran a t-test, please indicate if it was one- or two-sided, etc.).

*** As you used parametric tests in your study (i.e. t-tests), you should have first determined whether the data was normally distributed before selecting that test. In the stats section of the methods, please indicate how you tested for normality. If you did not test for normality, you must state something to the effect that "Data distribution was assumed to be normal but this was not formally tested."

5) Abstract and title:

The abstract should be no longer than 160 words and should communicate the significance of the paper for a general audience.

The title should be less than 100 characters including spaces. Make the title concise but accessible to a general readership.

6) Materials and methods:

*** Should be comprehensive and not simply reference a previous publication for details on how an experiment was performed. The text should not refer to methods "...as previously described."

Also, the materials and methods should be included with the main manuscript text and not in the supplementary materials.

7) Please be sure to provide the sequences for all your primers/oligos and RNAi constructs in the materials and methods.

*** You must also indicate in the methods the source, species, and catalog numbers (where appropriate) for all your antibodies. Please include species for all your antibodies.

8) Microscope image acquisition:

*** The following information must be provided about the acquisition and processing of images:

- a. Make and model of microscope
- b. Type, magnification, and numerical aperture of the objective lenses
- c. Temperature
- d. imaging medium
- e. Fluorochromes
- f. Camera make and model
- g. Acquisition software
- h. Any software used for image processing subsequent to data acquisition. Please include details and types of operations involved (e.g., type of deconvolution, 3D reconstitutions, surface or volume rendering, gamma adjustments, etc.).

10) Supplemental materials:

There are strict limits on the allowable amount of supplemental data. Reports may have up to 3 supplemental figures. There is no limit for supplemental tables. Currently, you have 4 supplemental figures, but we can give a bit of extra space in this case.

*** Please note that supplemental figures and tables should be provided as individual, editable files.

*** A summary of all supplemental material should appear at the end of the Materials and Methods section (please see any recent JCB paper for an example of this summary).

11) eTOC summary:

*** A ~40-50 word summary that describes the context and significance of the findings for a general readership should be included on the title page. The statement should be written in the present tense and refer to the work in the third person. It should begin with "First author name(s) et al..." to match our preferred style.

12) Conflict of interest statement:

JCB requires inclusion of a statement in the acknowledgements regarding competing financial interests. If no competing financial interests exist, please include the following statement: "The authors declare no competing financial interests."

13) A separate author contribution section is required following the Acknowledgments in all research manuscripts.

*** All authors should be mentioned and designated by their first and middle initials and full surnames and the CRediT nomenclature is encouraged (<https://casrai.org/credit/>).

14) ORCID IDs: ORCID IDs are unique identifiers allowing researchers to create a record of their various scholarly contributions in a single place. At resubmission of your final files, please consider providing an ORCID ID for as many contributing authors as possible.

15) Materials and data sharing:

All animal and human studies must be conducted in compliance with relevant local guidelines, such as the US Department of Health and Human Services Guide for the Care and Use of Laboratory Animals or MRC guidelines, and must be approved by the authors' Institutional Review Board(s). A statement to this effect with the name of the approving IRB(s) must be included in the Materials and Methods section.

*** As a condition of publication, authors must make protocols and unique materials (including, but not limited to, cloned DNAs; antibodies; bacterial, animal, or plant cells; and viruses) described in our published articles freely available upon request by researchers, who may use them in their own laboratory only. All materials must be made available on request and without undue delay. Please, indicate whether the fly strains and reagents generated in this study have been deposited in public repositories. If not, please state that they would be made available to the scientific community upon request in the 'Data availability' section.

All datasets included in the manuscript must be available from the date of online publication, and the source code for all custom computational methods, apart from commercial software programs, must be made available either in a publicly available database or as supplemental materials hosted on the journal website. Numerous resources exist for data storage and sharing (see Data Deposition: <https://rupress.org/jcb/pages/data-deposition>), and you should choose the most appropriate venue based on your data type and/or community standard. If no appropriate specific database exists, please deposit your data to an appropriate publicly available database.

16) Please note that JCB now requires authors to submit Source Data used to generate figures containing gels and Western blots with all revised manuscripts. This Source Data consists of fully uncropped and unprocessed images for each gel/blot displayed in the main and supplemental figures. The Source Data files will be directly linked to specific figures in the published article.

Since your paper includes cropped gel and/or blot images, please be sure to provide one Source Data file for each figure that contains gels and/or blots along with your revised manuscript files. File names for Source Data figures should be alphanumeric without any spaces or special characters (i.e., SourceDataF#, where F# refers to the associated main figure number or SourceDataFS# for those associated with Supplementary figures). The lanes of the gels/blots should be labeled as they are in the associated figure, the place where cropping was applied should be marked (with a box), and molecular weight/size standards should be labeled wherever possible.

B. FINAL FILES:

Thank you for this interesting contribution, we look forward to publishing your paper in Journal of Cell Biology.

Sincerely,

Jodi Nunnari, Ph.D.
Editor-in-Chief
The Journal of Cell Biology

Lucia Morgado-Palacin, PhD
Scientific Editor
Journal of Cell Biology

Reviewer #1 (Comments to the Authors (Required)):

The authors have adequately addressed all of my major concerns in the revised manuscript. I am now convinced that REC does localize to the mitochondrial matrix and promotes mtDNA recombination. This is an excellent paper that uncovers the molecular machinery of mtDNA recombination and suggests potential function of mtDNA recombination system in mitochondrial genome maintenance.

Reviewer #3 (Comments to the Authors (Required)):

I would like to thank the authors for addressing the points I -- and the other reviewers -- raised and for providing a constructive discussion where new experimental data was not presented. I find this submission to be of much higher quality in both the contents and the presented data. I recommend the manuscript to be accepted as is.